# Subclonal mutation selection in mouse lymphomagenesis identifies known cancer loci and suggests novel candidates

Philip Webster[1,2,3], Joanna C. Dawes[1,2], Hamlata Dewchand[1,2], Katalin Takacs[1,2], Barbara Iadarola[1,2], Bruce J. Bolt [1,2], Juan J. Caceres [4], Jakub Kaczor[1,2], Gopuraja Dharmalingam[1,2], Marian Dore[1,2], Laurence Game[1,2], Thomas Adejumo[1,2], James Elliott[1,2], Kikkeri Naresh[3], Mohammad Karimi[1,2], Katerina Rekopoulou[1,2], Ge Tan[1,2], Alberto Paccanaro [4] & Anthony G. Uren [1,2]

Determining whether recurrent but rare cancer mutations are bona fide driver mutations remains a bottleneck in cancer research. Here we present the most comprehensive analysis of murine leukemia virus-driven lymphomagenesis produced to date, sequencing 700,000 mutations from >500 malignancies collected at time points throughout tumor development. This scale of data allows novel statistical approaches for identifying selected mutations and yields a high-resolution, genome-wide map of the selective forces surrounding cancer gene loci. We also demonstrate negative selection of mutations that may be deleterious to tumor development indicating novel avenues for therapy. Screening of two BCL2 transgenic models confirmed known drivers of human non-Hodgkin lymphoma, and implicates novel candidates including modifiers of immunosurveillance and MHC loci. Correlating mutations with genotypic and phenotypic features independently of local variance in mutation density also provides support for weakly evidenced cancer genes. An online resource http://mulvdb.org allows customized queries of the entire dataset.

[1] MRC London Institute of Medical Sciences (LMS), Du Cane Road, London W12 0NN, UK. [2] Institute of Clinical Sciences (ICS), Faculty of Medicine, Imperial College London, Du Cane Road, London W12 0NN, UK. [3] Imperial College Healthcare NHS Trust, London W12 0HS, UK. [4] Centre for Systems and Synthetic Biology, Department of Computer Science, Royal Holloway, University of London, Egham TW20 0EX, UK. These authors contributed equally: Philip Webster, Joanna C. Dawes  Correspondence and requests for materials should be addressed to A.G.U.(email: anthonyuren@gmail.com)

  1

Increasing cohort sizes of human tumor sequencing has revealed large numbers of rare clonal mutations, the contribution of which is difficult to prove due to a lack of statistical power, giving rise to false positives and negatives[1]. It is similarly challenging to determine how non-coding mutations, large-scale copy number alterations, and epigenetic mechanisms contribute to disease. The fraction of rare and non-coding mutations that drive cancer is largely unknowable. The data available to identify cancer drivers from tumor sequencing studies could be increased through the inclusion of subclonal mutations in both premalignant samples as well as mature tumors; however, this requires numbers sufficient to demonstrate that the early stages of selection have taken place. In this study, we use somatic insertional mutagenesis in mice as a model to demonstrate that low abundance mutations that are only rarely found as clonal mutations in advanced-stage disease can be effectively employed to identify known cancer drivers and differentiate rare disease-causing mutations from passenger mutations.

Murine leukemia virus (MuLV)-induced lymphoma is an ideal model to study selection of subclonal mutations. Cloning integration mutations by ligation-mediated PCR requires a fraction of the sequencing coverage needed to identify other mutation types, allowing large numbers of integration mutations to be identified with unparalleled sensitivity. Furthermore, gamma retroviruses are not subject to remobilization, can integrate in any sequence context, and localized bias of the orientation of integrations can be used as a measure of selection that is independent of regional variation in integration density[2].

Infection of newborn mice with MuLV causes a systemic life-long viremia whereby viral integrations deregulate and truncate nearby genes by diverse mechanisms, eventually causing hematologic malignancies[3]. A high proportion of the recurrently mutated loci correspond to known drivers of human malignancies[3,4]. Historically, these screens focused on mutations present in clonal outgrowths as evidence of their role in malignancy; however, recent pyrosequencing of MuLV lymphomas has also shown selection taking place within subclonal populations of cells[2].

Using a novel insertion site cloning protocol, that is able to detect subclonal retroviral integrations with unprecedented sensitivity, we cloned more than 3000 clonal and 700,000 subclonal mutations across a spectrum of >500 MuLV-induced T cell and B cell lymphoid malignancies from two BCL2 transgenic models over a time course of lymphomagenesis. From these we find both positive and negative selection of insertions throughout all stages of lymphomagenesis, and that in late-stage disease both clonal and subclonal populations identify more than 100 known cancer drivers and regions implicated in non-Hodgkin lymphoma (NHL) by coding mutations, copy number aberrations, and genome-wide association studies (GWAS). This resource can be used to prioritize rare but recurrent mutations from human tumors for further study.

## Results

### An MuLV time course quantifies the transition to lymphoma.
To observe mutation selection during lymphomagenesis we generated a diverse set of B cell and T cell-derived lymphoid malignancies, sacrificing animals with advanced-stage disease, as well as over a time series prior to disease development. Moloney MuLV typically results in a T cell leukemia/lymphoma; however, subtype and mutation profile can be skewed by genetic background and predisposing germline alleles[5,6]. To generate a diverse spectrum of B and T cell malignancies we infected newborns on two genetic backgrounds using two hBCL2 expressing transgenic models, Vav-BCL2 (ref. [7]) and Emu-BCL2-22 (ref. [8]). The

t(14;18)(q32;q21) IGH/BCL2 translocation drives enforced expression of the antiapoptotic protein BCL2 and is one of the earliest and most common initiating mutations of follicular lymphoma (FL) and diffuse large B cell lymphoma (DLBCL). Overexpression of BCL2 is also frequently observed in B cell chronic lymphocytic leukemia (CLL).

All mice developed lymphoid malignancies with latency ranging 42–300 days (Fig. 1a–c), with enlarged spleens, thymuses, and lymph nodes observed in all cohorts. Disease onset was significantly accelerated by the Vav-BCL2 transgene ($p = 0.0001$) (Fig. 1a) and by the Emu-BCL2-22 transgene on a C57BL/6 background ($p = 0.0163$) (Fig. 1c) compared with their littermate controls. The $F_1$ background developed lymphoma more rapidly than equivalent C57BL/6 cohorts (Supplementary Figure 1).

Immunophenotyping of spleen cell suspensions of 345 animals demonstrated variable B and T cell proportions in all cohorts, reflecting the broad tropism of Moloney MuLV (Fig. 1d–f, gating strategy outlined in Supplementary Fig. 2a). BCL2 transgenic cohorts yielded a higher proportion of CD19+ B cell lymphomas compared to wild-type mice (Fig. 1g), most notably in the Vav-BCL2 cohort. Spleen suspensions segregate into two groups, with either a majority of T cells or of B cells (Fig. 1h). T cell lymphomas were primarily CD4+, less frequently CD4−CD8− and rarely CD8+ or CD4+CD8+ (Supplementary Fig. 2b). B cell lymphomas were generally immunoglobulin light chain positive indicating a mature B cell phenotype (Supplementary Fig. 2c). MuLV-infected Vav-BCL2 transgenic mice displayed a disproportionate outgrowth of PNA+ CD95+ germinal center B cells, and isotype switching to IgG as has been previously described in this strain[9] (Supplementary Fig. 2c, d).

To observe rates of selection of mutations in animals at all disease stages we also generated cohorts sacrificed at days 9, 14, 28, 56, 84, and 128 post-infection and harvested spleens (Supplementary Table 1). QPCR of virus transcript and copy number indicated that virus replication was detectable at day 9 and reached saturation at day 14 (Fig. 2a, b). This suggests a high proportion of mutagenesis occurs in the first 14 days post-infection, with subsequent rare events of superinfection and selective pressure shaping the mutation profile and the eventual clonal outgrowth of late-stage lymphomas.

Retroviral integration sites from all animals were identified using a novel Illumina HiSeq based protocol (a revision of methods described in Koudijs et al.[10] and Uren et al.[11] and summarized in Supplementary Figure 3 and the Methods section). The protocol switches strands of the standard Illumina adapter to prevent any non-MuLV library fragments from amplifying and reduces the PCR cycles to 25 divided over two nested steps. We sequenced libraries from the lymphoid organs of 355 diseased animals, 166 animals sacrificed at predetermined time points, and from control DNAs (human and uninfected mouse DNA as a measure of PCR artifacts). Sequencing these libraries identified more than 700,000 unique integration sites, the vast majority of which are represented by a single read, suggesting the vast majority of integrations have not undergone clonal expansion and that some may only represent a single DNA molecule in the original sample.

MuLV generates tumors with 100% penetrance, resulting from independent competing clones and/or related subclones within each animal. We sought to distinguish the early stages of disease prior to clonal expansion from rapidly dividing samples with clonal outgrowth by using the relative abundance of mutations in each sample. The relative clonality of integrations within each ligation was estimated using the number of individual sheared DNA fragments identified for each insert. A single clonal outgrowth of pure tumor cells containing few mutations will yield high coverage of each mutation, whereas a clonal outgrowth

with dozens of concurrent mutations alongside a large proportion of non-tumor DNA will yield low coverage for even the most clonal mutation. For comparison between samples we generated normalized clonality values (NC values) where the most clonal integration within each sample was normalized to a value of 1 (Fig. 2c).

**Quantifying lymphoma progression by clonal outgrowth.** In Fig. 2c the 50 insertions with the highest NC values within each sample are ordered by their relative abundance and plotted as bar graphs. Early-stage samples from the time course had a flat profile of subclonal mutations with the majority represented by a single read/DNA fragment, whereas later stage lymphoma samples are

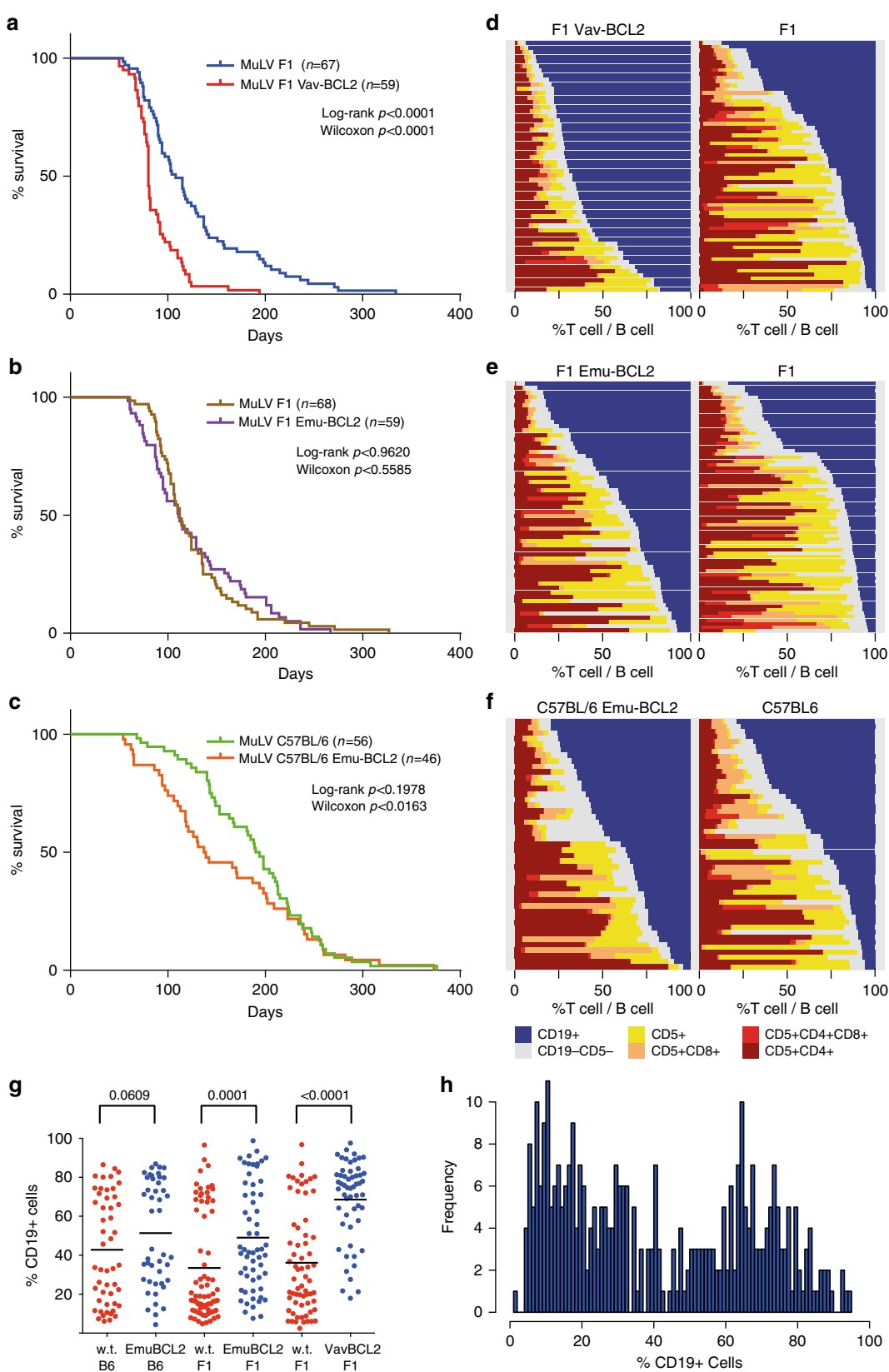

dominated by outgrowth of up to 20 clonal integrations. Entropy was employed as a description of the clonality of integrations within each sample (based on the prior use of the Shannon entropy to estimate clonal outgrowth of T lymphoma[12] and in mathematical models of leukemia[13]). Entropy calculations from the 50 most clonal integrations yielded high scores for early-stage samples and low scores for advanced-stage lymphoma.

To verify whether these clonality values for each insert were reproducible, a series of test libraries were generated from two spleen DNAs. The majority of mutations with NC above 0.1 were found within all six libraries from the same DNA sample prepared on four separate occasions (Supplementary Figs. 4–7). This was truer for a lower entropy library (average entropy $E =$ 3.29) with four distinct clonal integrations than a higher entropy library (average entropy $E = 3.56$) with a more continuous distribution of clonality. In general reproducibility is a function of clonality and these results suggest clonal integrations can be distinguished from subclonal integrations with some consistency, although it cannot be ruled out that some integrations will be underrepresented due to amplification biases and other artefacts of library construction. We also generated a dilution series of two spleen DNAs to demonstrate that clonal integrations can be reproducibly detected when diluted 100-fold into a second DNA sample (Supplementary Fig. 8) and that abundance is a function of initial concentration. Results from these experiments are discussed in detail in the legends of Supplementary Figures 4–8.

Entropy scores reduce the information of a clonality profile to a single number and this number can vary between libraries of a very similar shape (e.g. Supplementary Fig. 4a). As an independent complementary approach, we used distance measures to cluster clonality profiles by their shape. Both Dynamic Time Warping[14] (DTW) and the Kolmogorov–Smirnov statistic were used to measure the difference in shape between all insert profiles and a distance matrix was constructed. Clustering on the DTW matrix yields two clusters (Fig. 3a) that place the majority of early-stage time course samples within the cluster with higher entropy values >3.5 (Fig. 3b, c). These groups differed by only 19 of 512 samples when using the Kolmogorov–Smirnov statistic (Supplementary Data 1). Throughout the time course, entropy values remain high until days 56 and 84 when an increasing fraction of samples display lower values indicating clonal outgrowth (Fig. 3d). Overall, clonal outgrowth is a function of mouse age and strongly correlates with symptomatic disease. Diseased mice (i.e. those sacrificed due to symptoms) with entropy >3.5 likely represent animals where lymphoma cells had not disseminated to the spleen, but had symptoms arising from other organs (thymus, lymph nodes). For subsequent analyses, we define late-stage samples as those with clonal outgrowth with a low entropy and use a cut-off of entropy <3.5 and NC >0.1 to define late-stage clonal insertions (Fig. 3e, f).

**Kinetics of mutation selection throughout a time course.** MuLV has integration biases that vary substantially throughout the genome such that mutation density is insufficient to differentiate selected driver mutations from passengers. For this reason, we assessed selection over the time course to define driver mutations. First, by limiting analysis to the 3051 clonal integrations of the late-stage lymphomas we identified 311 common integration sites (CISs) by Gaussian kernel convolution[15] (GKC), i.e., by estimating the smoothed density distribution of integrations over the genome compared to random distributions (Supplementary Data 2). Candidate genes were automatically assigned using the KCRBM R package[16]. Examining all insertions within 100kb windows of clonal CIS peaks over all time points demonstrated a gradual increase in the proportion of inserts at these loci from day 9 through to late-stage lymphoma samples (Supplementary Fig. 9a–c).

To quantify the significance of this selection we used contingency table tests (Fisher's exact) to compare the number of integrations in windows surrounding loci in early-stage mutations (days 9 and 14), late-stage clonal, and late-stage subclonal mutations. Ranking loci using the exact test comparisons between early- and late-stage mutations yields similar results using either subclonal or clonal mutations (Supplementary Fig. 9d, Supplementary Data 2).

$p$-Values for all early/late, clonal/subclonal Fisher's exact tests for the top 50 clonal CIS loci are illustrated in Fig. 4 in the blue heat map. In some cases, high ranking clonal CIS loci demonstrate weak selection between early mutations and late-stage clonal mutations (e.g. Bzrap, Rreb1), suggesting these are more likely to be passenger mutations resulting from integration site biases of MuLV. Late-stage subclonal mutations outnumber clonal mutations by 100-fold. Including these subclonal mutations in the analyses, i.e., comparing all late-stage integrations to early mutations, reveals some CIS loci with selection that is more significant than in clonal analysis alone (Supplementary Data 2), including verified human cancer genes such as REL, EBF1, ERG, ELF4, MYCL, KIT, and KDR. The finding of known cancer drivers at these loci demonstrates that analysis of selection over a time course offers an enlarged dataset from which to identify previously validated cancer drivers and by extension, potentially identify novel genes not previously implicated in disease.

We also considered other criteria as evidence for selection. A recent study of MuLV-induced T cell lymphomas used orientation of integrations as evidence that there is selection for deregulation of nearby genes[2], i.e., loci bearing insertions that are decidedly biased in one direction are likely to have undergone selection for their effects on a nearby gene that would differ if the insertion orientation were opposite.

The red heat map of Fig. 4 indicates that the majority of the top 50 clonal CIS loci also have a significant bias for integrations on one strand or the other. We additionally observed phenotypic bias (selection specific to B cell or T cell lymphoma, the yellow heat map) and genotypic bias (selection in cooperation with the BCL2 transgenes, the green heat map). The majority of the top 50 clonal CIS loci demonstrate biases of strand specificity, lymphoma subtype, and/or genotype specificity. Importantly there is substantial overlap between all four selection criteria (stage, orientation, immunophenotype, and genotype), suggesting these criteria can be used in concert to provide corroborating evidence for selection.

**Fig. 1** Variable latency and immunophenotype of MuLV lymphoma from wild type and BCL2 transgenic mice. **a** The Vav-BCL2 transgene significantly reduced latency on an F₁ background. **b**, **c** The Emu-BCL2-22 transgene significantly reduces latency on a C57BL/6 background but not F₁ background. The Emu-Bcl2-22 C57BL/6 cohort had a significantly shorter latency than wild-type C57BL/6 controls and both C57BL/6 cohorts had longer latency compared with F1 equivalents (Supplementary Fig. 1). **d–f** Stacked bar charts on the right represent the immunophenotyping of spleen suspensions from each cohort. Each row represents one spleen. Colors in each row represent the proportion of B cells (blue CD19+) and T cells (yellow CD5+ CD4– CD8–, light orange CD5+ CD8+, dark orange CD5+ CD4+ CD8+, and red CD5+ CD4+) in each sample. BCL2 transgenes increase the proportion of B cells in all cohorts and the mixture of T cell lymphoma subtypes is highly variable. **g** The proportion of CD19+ B cells is increased by both BCL2 transgenes. **h** Histogram of all CD19+ proportions from all cohorts combined is a bimodal distribution that can be segregated into those consisting primarily of B cells (>50%) and T cells

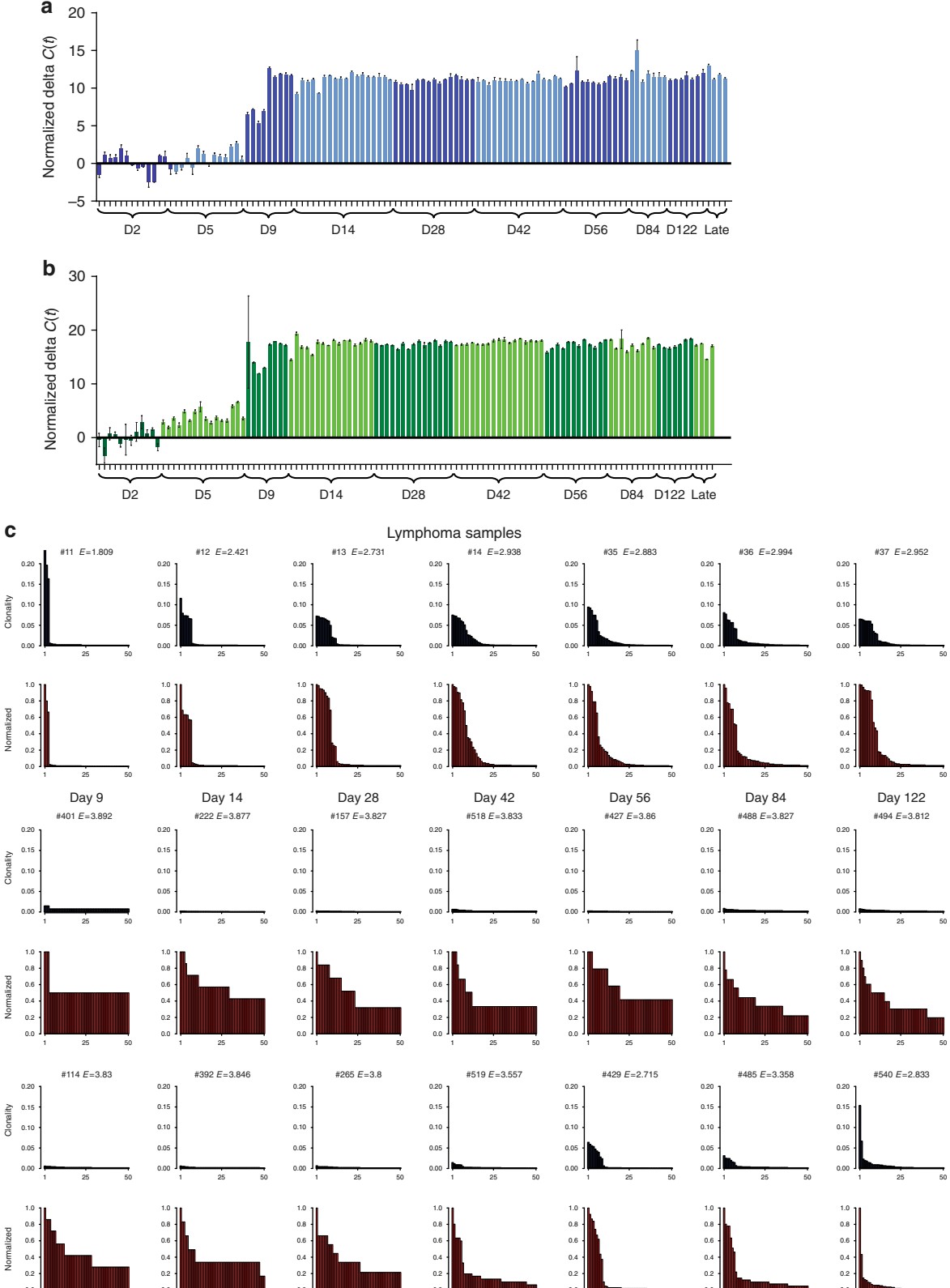

**Fig. 2** Quantifying the progression of MuLV replication and clonal outgrowth of resulting lymphoma. Virus copy number and expression level was quantified by QPCR of genomic DNA (**a**) and RTQPCR of cDNA (**b**) extracted from spleen samples of time course animals. Error bars represent s.d. of 3 technical replicates per DNA/RNA sample. (**c**) Profiles of the relative abundance of the top 50 most clonal integrations from a cross section of mature lymphoma (upper 2 rows) and time course samples (lower 4 rows) are represented as bar graphs. Non-adjusted clonality is indicated in blue, normalized clonality (such that the most abundant integration has a value of 1) are the graphs in red. Asymptomatic animals from early time points display a relatively flat profile whereas later time points and mice with symptomatic lymphoma show clear signs of clonal outgrowth. Shannon entropy values (*E*) are displayed on each graph

**Identifying loci undergoing selection genome wide**. The gold standard for defining driver mutations is not only clonal outgrowth, but that the profile of mutations observed is significantly skewed from the profile of background passenger mutations. Rarity of clonal mutations within a region does not, however, rule out selection in that region. Of the 300,000 integrations from late-stage diseased animals, only a fraction are located within the

regions surrounding clonal CIS loci. To see if selection is observed in regions outside clonal mutation CIS loci, we extended analysis across the entire genome using both 10kb tiling and 100kb sliding windows at 10kb intervals.

Examining the entire genome reveals significant local biases for early versus late stage, strand bias, genotype, and immuno-phenotype (B cell/T cell lymphoma). After multiple testing

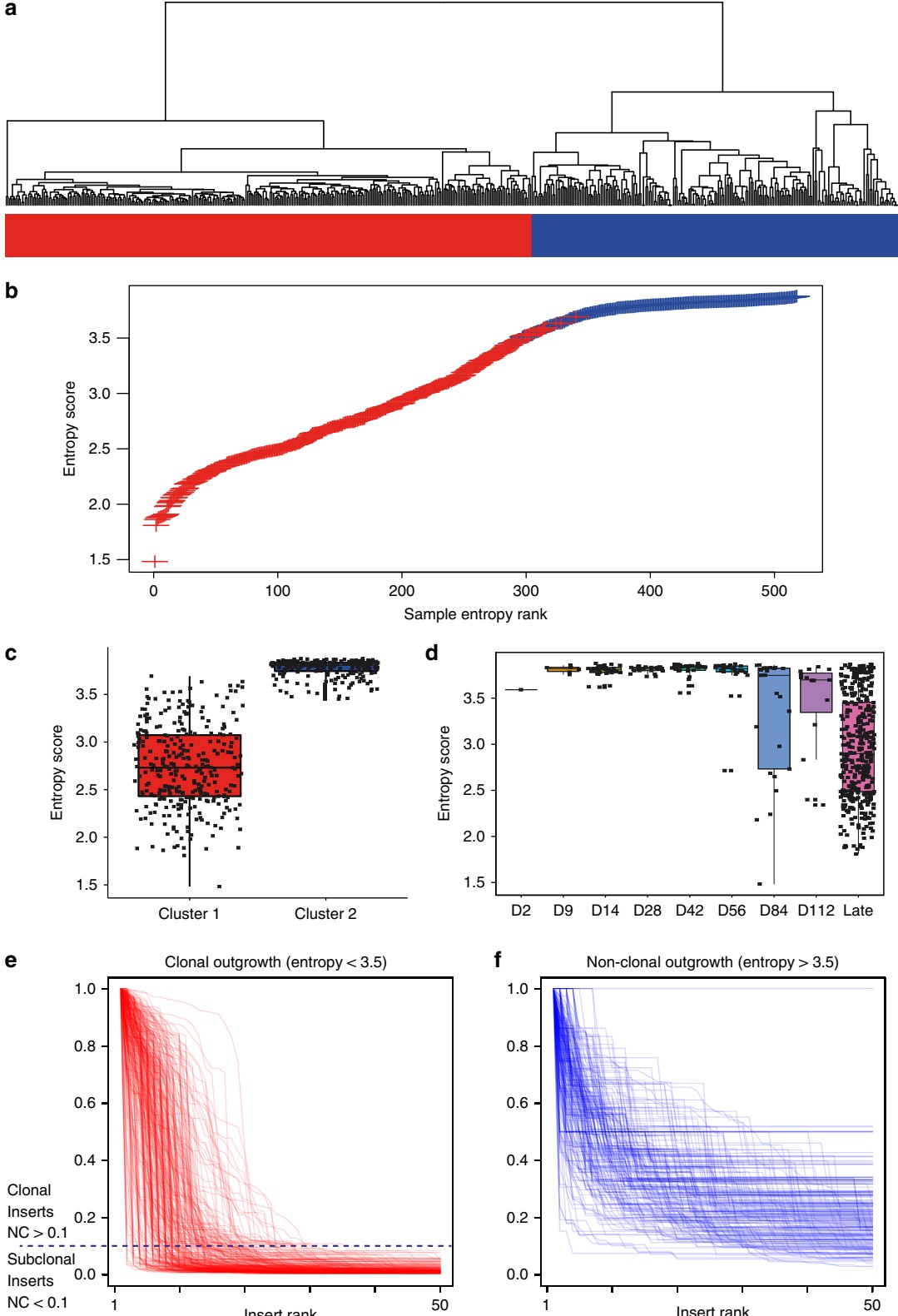

correction, we identified 174 late-stage specific loci with a false discovery rate (f.d.r.) below 0.05, including dozens of windows containing only subclonal insertions. (Supplementary Fig. 10 and Supplementary Data 3). The Venn diagram in Fig. 5a illustrates that there is substantial overlap of late-stage selected loci with equivalent loci found to be strand specific (49 loci), and genotype specific (37 *BCL2* loci, 15 wild-type loci). This is also true for immunophenotype specific loci (19 B cell loci, 11 T cell loci). Although biases were observed for some loci between males and females, none were found to be significant after multiple testing correction.

In regions of high insert density subclonal mutations form a high-resolution map of the selective pressures surrounding known oncogenes. The distribution of integrations in different subsets of samples is illustrated across a single chromosome (chr15) in Supplementary Figure 1a. Tracks representing inserts and the relative levels of selection across the genome (calculated using Fisher's exact test) indicates extensive selection outside regions identified by GKC clonal inserts. The central region of chromosome 15 (chr15:62,000,000–63,000,000) with the highest concentration of late-stage integrations is the *Myc/Pvt1* locus (Fig. 5b). The surrounding region harbors multiple clusters of selection spanning from upstream of *Myc* and extending through multiple clusters downstream as far as the *Gsdmc* gene family locus (Supplementary Fig. 11b). This distribution of selected mutations over a 2Mb region concurs with the recent finding that copy number gains of the entire segment, incorporating *Myc*, *Pvt1*, and the *Gsdmc* family locus, is required to give acceleration of cancer in mouse models[17].

Orientation or strand bias is a unique criterion, in that it is independent of the integration biases of MuLV that may be influenced by cell type or genotype. To validate that orientation bias is indeed a function of selection we calculated bias using equal numbers of integrations from early- and late-stage cohorts (i.e. 80,000 integrations). No loci from the early-stage inserts are significant after multiple testing correction, but the late-stage insert subset identifies 16 loci. This illustrates that the increased significance of strand bias in the late-stage cohort is not merely a function of greater statistical power from larger number of integrations, but rather evidence of selection for integrations that deregulate or disrupt genes (Supplementary Data 4).

**Selection effectively identifies known human cancer drivers.** Supplementary Data 5 lists all candidate genes associated with one or more of our selection criteria. Supplementary Data 6 lists the subset of these loci corresponding to genes from the cancer gene census[18]; in total 47 genes located at 43 loci. Of the 47 genes, 27 map within 200,000kb of a clonal CIS with a *p*-value <0.05 (Fisher's exact test); however, an additional 21 genes at 20 loci are implicated by subclonal selection criteria that were not identified by clonal CIS demonstrating subclonal mutations can provide additional statistical evidence to implicate cancer drivers for loci lacking sufficient clonal mutations to make this determination.

We also compared the list of candidate genes identified by any criteria with a set of 12 cohorts of hematological malignancies present in the cBio portal[19]. All protein coding candidates generated by KCRBM (without curation) or the curated candidate gene lists were used to identify human orthologues using BioMart (http://www.ensembl.org/biomart/martview/). The set of 78 MuLV candidate genes found mutated in two or more samples from any study is depicted in Fig. 6a. For the majority of cohorts we find significant overlap between either the KCRBM candidates or the curated genes (Fig. 6b and Supplementary Data 7). We see most overlap with a pan NHL study (consisting of DLBCL and FL) and cohorts of mature B cell-derived lymphoma (DLBCL, MM, MCL). Importantly the overlap is also significant when examining the set of genes mutated only once in each study, i.e., the set of most rarely mutated genes from most cohorts overlaps significantly with the candidate lists, demonstrating this dataset can be used as corroborating evidence for rarely mutated genes in human sequencing cohorts.

Across the set of genes identified there is a notable prevalence for genes that are known to be deregulated by translocations and/ or copy number aberrations as well as those found in GWAS in humans. Using a list of 278 selected regions identified in this screen by any criteria, 273 were mapped unambiguously to an orthologous region on Hg19. We overlapped this set of loci with focal copy number aberrations of five human studies of mature B cell lymphoma and found a significant degree of enrichment in four of the five datasets (Supplementary Table 2).

Recurrent large-scale copy number changes in human B NHL suggest the involvement of multiple genes within these regions. We find corresponding loci where selection is evident over large regions incorporating multiple genes. Aside from the above-mentioned *Myc/Pvt1/Gsdmc* family locus (Supplementary Fig. 11b), we see selected regions surrounding the *Rel/Bcl11a* locus (orthologous to human 2p12–16 amplicons of CLL and DLBCL Supplementary Fig. 11c), the *Slamf* gene family which regulate lymphocyte survival, activation, and co-stimulation (orthologous to human 1q21–23 amplicons of multiple myeloma and DLBCL Supplementary Fig. 11d), and the *Gimap* gene family of GTPases that also regulate lymphocyte survival and development (orthologous to amplicons of the distal arm of human 7q seen in FL, DLBCL, and Burkitt lymphoma) (Supplementary Fig. 11e). We also see selected loci spanning the region surrounding *Prdm1* (orthologous to deletions of human 6q21 in B cell NHL and other hematologic malignancies Supplementary Fig. 11f).

The use of *BCL2* transgenic animals expands the scope of mutations identified beyond loci typically identified by MuLV in wild-type animals. Of 37 loci that are *BCL2* transgene specific, and 19 loci that are B cell specific, we find selected regions near known B cell lymphoma and leukemia drivers, including *Pou2f2*, *Ebf1*, *Ikzf3*, and *Bcl6*. The most specific locus for both *BCL2* transgenic animals and for B cell lymphoma is *Pou2f2* (Supplementary Fig. 11g) which is recurrently mutated in human FL and DLBCL. Recurrent missense mutations reduce Pou2f2

**Fig. 3** Using distance-based measures and entropy as indicators of clonal outgrowth of lymphoma. **a** Dynamic Time Warping was used to cluster clonality profiles of all samples and identifies two major groups; early-stage samples (blue) and samples undergoing clonal outgrowth (red). Near identical clusters were obtained using the Kolgomorov–Smirnov statistic (Supplementary Data 1). **b** Samples are plotted comparing entropy score by rank and individual samples are colored by cluster branch, indicating both entropy scores and clustering give a similar bifurcation of samples. **c** Distribution of entropy scores between the two clusters indicates an entropy value of 3.5 effectively separates the groups. The mean (horizontal line), ±1 s.d. (box), and ±2 s.d. (vertical line) are indicated. **d** Distribution of entropy scores between different time points indicates a progressive increase in the frequency of clonal outgrowth (mean (horizontal line), ±1 s.d. (box), and ±2 s.d. (vertical line)). Superimposing the clonality profiles of all samples within each cluster indicates consistent shape within the low entropy group (**e**) and within the high entropy group (**f**). A normalized clonality value of 0.1 is used to differentiate clonal and subclonal mutations within the late-stage clonal outgrowth samples

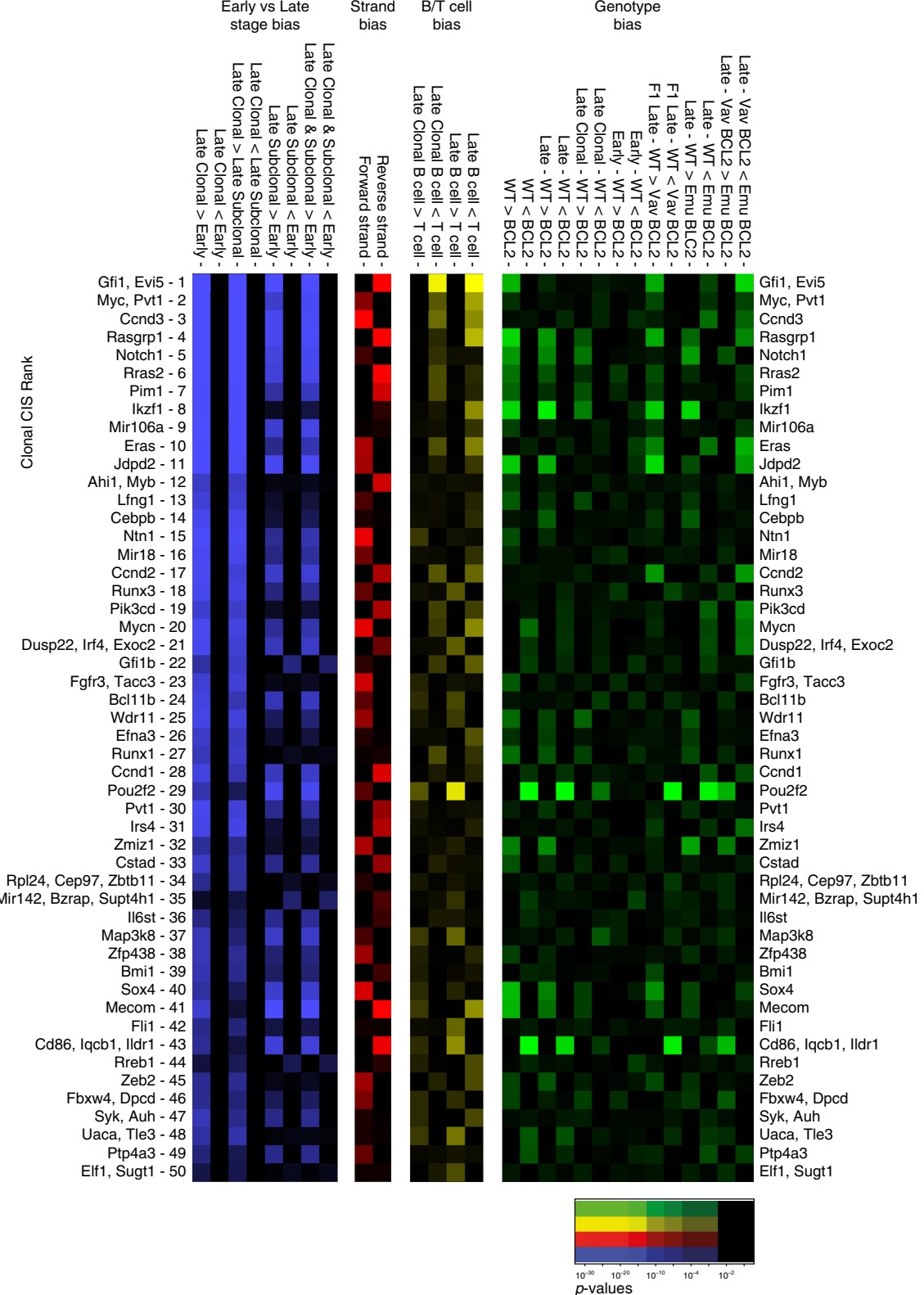

**Fig. 4** Multiple criteria indicate selection of both clonal and subclonal mutations at CIS loci. Four heat maps representing the relative levels of selection observed between different categories of integrations. Fisher's exact tests were performed counting the inserts within 100kb windows surrounding each of the top 50 clonal insert loci. Blue indicates comparisons between early and late-stage integrations. Red represents integration orientation bias (forward or reverse strand). Yellow represents specificity for B cell (>50% CD19) versus T cell lymphomas. Green represents specificity between different genotypes. *p*-Values for Fisher's exact tests are indicated by color intensity

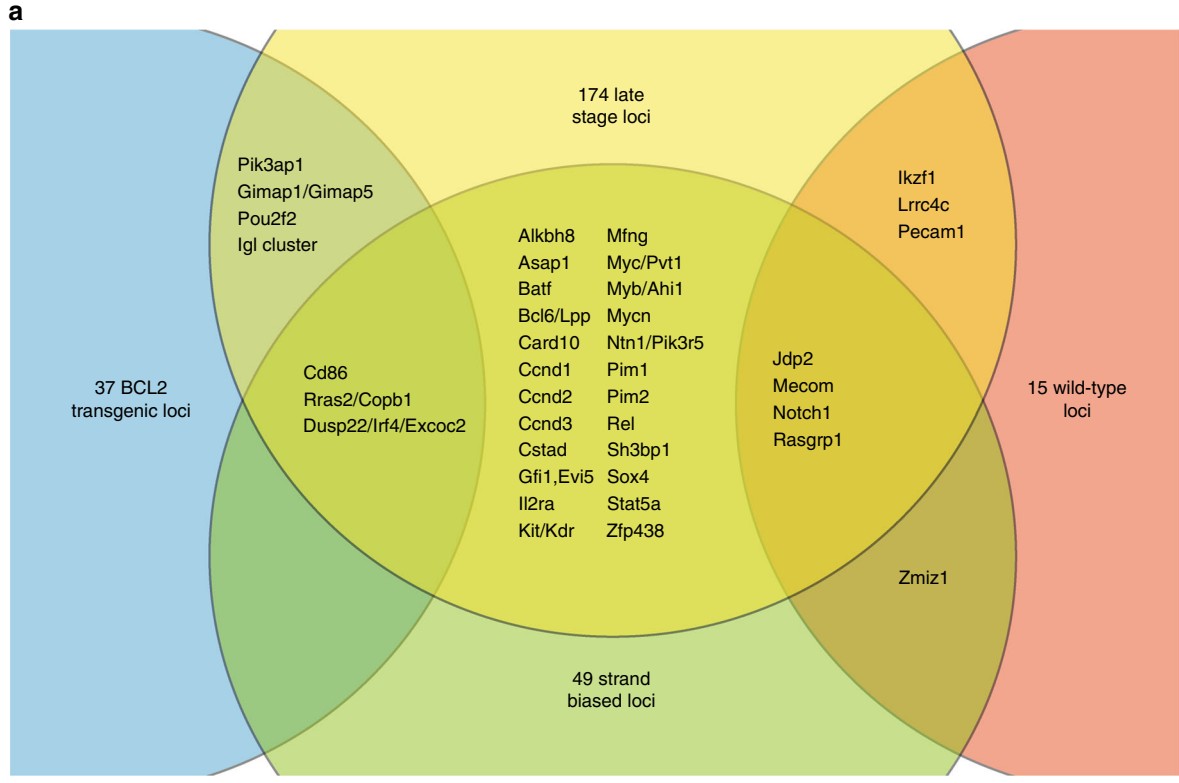

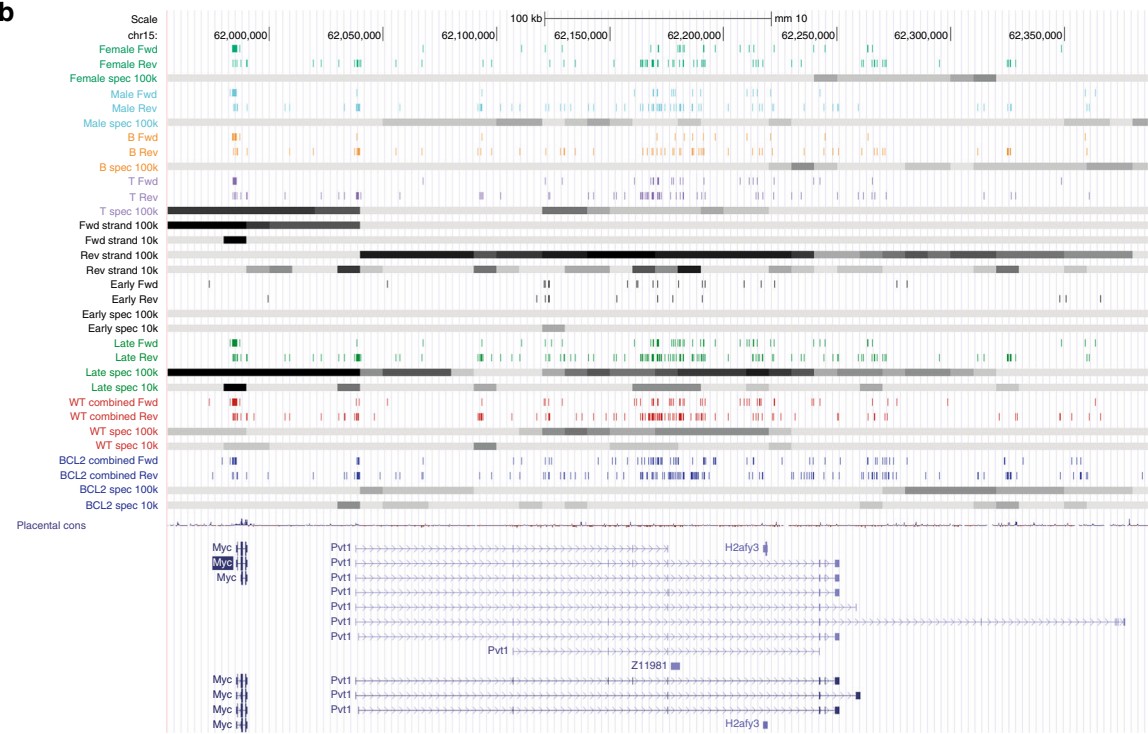

**Fig. 5** Genome-wide scanning of subclonal mutation distributions identifies regions undergoing selection. **a** Genome-wide contingency table tests of all mutations identifies loci that are late-stage specific, strand biased and genotype biased. The Venn diagram demonstrates substantial overlap between loci identified by these criteria. **b** Distribution of integrations over the *Myc/Pvt1* locus. Each row of colored vertical lines represents the forward and reverse strand integrations of each category of mice. Gray bands below each colored row represent the level of selection evidenced by contingency table tests. Late-stage specific integrations are evident throughout the region; however, integrations upstream of *Myc* are primarily on the forward strand and T cell specific whereas integrations within the *Pvt1* gene are in the reverse orientation and somewhat biased toward wild-type mice

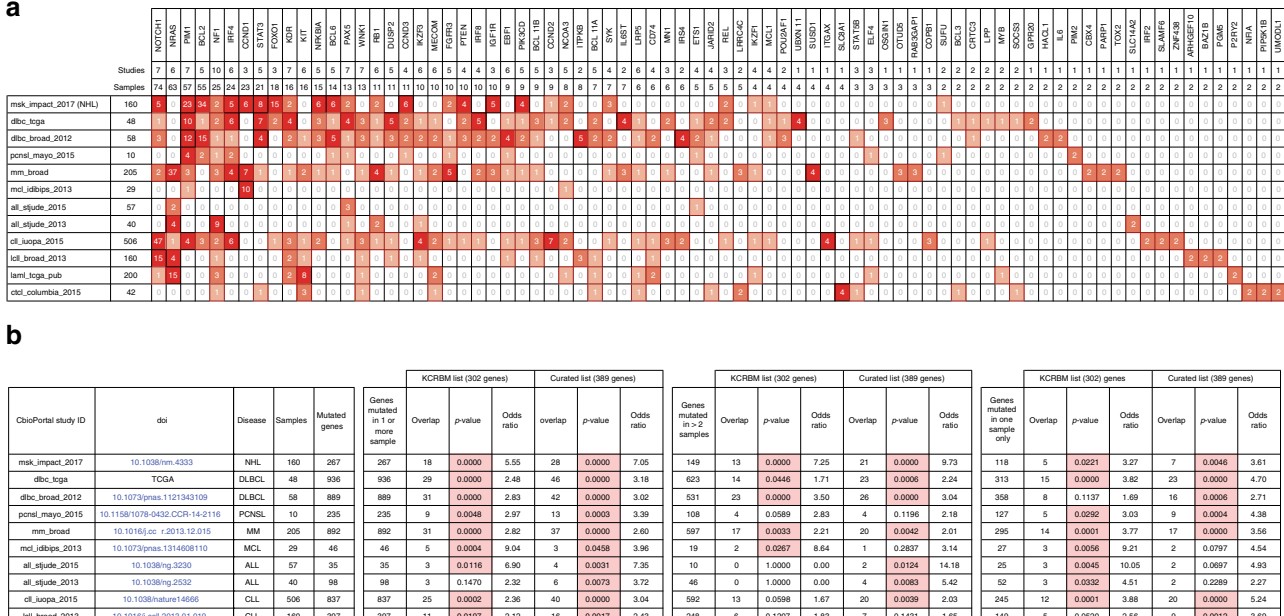

**Fig. 6** Overlap of CIS loci with exome sequencing studies of hematologic malignancies. Human orthologues were identified for all candidate genes (an automated list KCRBM, and a curated list) using biomart and compared to lists of genes with coding mutations in 12 cohorts of hematologic malignancy in cBio portal. **a** All genes found mutated in at least two samples over all cohorts are listed with mutation counts from each cohort. The full overlap for all cohorts is listed in Supplementary Data 7. **b** The significance of overlap between the set of candidate orthologues and the set of mutated genes in each study is calculated using a Fisher's exact test

transactivation activity and B lymphoma cell lines expressing these have a survival advantage[20], conversely DLBCL cell lines appear to be addicted to *POU2F2* expression[21]. The position of these insertions within several known tumor suppressor genes is suggestive of disruption. Similar intragenic distributions of integrations have previously been described for the tumor suppressor *Ikzf1* which is mutated or deleted in both B and T-ALL[22,23] (Supplementary Fig. 11h), and this pattern is also observed for the tumor suppressors *Ikzf3* and *Ebf1* (Supplementary Figures 11i, j), both of which have inactivating mutations in FL and DLBCL but more typically B-ALL[22,24–26]. *Cxxc5* (late-stage specific) also has a similar pattern of intragenic integrations (Supplementary Fig. 11k) and is deleted and epigenetically silenced in acute myeloid leukemia[27]. The majority of our insertions at the *Pou2f2* locus are intragenic, consistent with the tumor suppressor function observed in FL.

MuLV is known to deregulate genes via a diverse set of mechanisms including distal interactions of virus enhancers via 3D conformation of the genome[28]. The integration bias of MuLV for enhancers[29] and capacity for long-range interactions likely explains the clusters of late-stage/strand biased/genotype specific integrations observed around gene poor loci. To identify candidates most likely to cause disease, we searched for additional evidence in the literature supporting the role of candidate genes in hematologic malignancies, with an emphasis on data from human lymphoid malignancies and a particular focus on *BCL2*-driven B cell lymphoma. Supplementary Data 8 lists evidence for 194 genes from our list of candidates flanking selected loci. Some candidates are only implicated by one or two studies, and as such this dataset provides independent experimental corroboration. In addition to 47 genes identified in the cancer genome census we also find 30 genes identified in the Intogen mutational cancer driver gene list[30].

To identify genes that had their expression or transcript structure deregulated by insertions (and the mutation clonality required to observe this) we sequenced RNA samples from a subpanel of 26 BCL2 transgenic tumors. Isolated fusion transcripts were identified throughout the genome (Supplemental Data 9). The most frequent and clonal fusion transcripts identified were in the 3′ UTR of Mycn, causing upregulation of expression (Supplementary Fig. 12a, b and Supplementary Data 9) as observed in previous studies[31]. Another tumor had two intragenic fusions within Notch1, this tumor having the highest Notch1 expression of all 26 (Supplementary Fig. 12c, d). Insert clonality corresponds to transcript read numbers, with subclonal integrations (NC <0.1) in these tumors giving few if any detectable transcripts.

**Selection of loci implicated by GWAS of lymphoma.** In a previous study we found overlap between CIS loci and loci associated with familial CLL[6]. Supplementary Data 10 summarizes the literature of GWAS studies of ALL, FL, and DLBCL comprising 26 loci implicated in one or more studies. Of 278 loci from our study that were significant by any criteria 14 overlap with the set of 25 candidate genes (a significant enrichment by Fisher's exact test p < 0.0001). The *IKZF1* and *PIP4K2A/BMI1* loci are associated with ALL[32,33]. GWAS of mature B cell lymphomas have identified associations with FL (*LPP, HLA* loci, *PVT1, CXCR5, ETS1*, and *BCL2*) and with DLBCL (*LPP, EXOC2, HLA-B*, and *PVT1*)[34,35].

The second most specific locus for *BCL2* transgenic mice is *Cd86* (late-stage, strand bias, BCL2), which is suggestively implicated by two GWAS studies of FL and DLBCL[34–36]. We additionally find other loci encoding co-stimulatory/co-inhibitory signaling (Supplementary Figures 11l–p). The loci encoding *Cd86* ligands *Ctla4* and *Cd28* and their neighboring homolog *Icos* show late-stage-specific selection, and polymorphisms in this region have also been associated with various NHL subtypes[37]. The B7 family and their receptors are also implicated by insertions near *Icoslg* (late stage) and the *Cd274* (Pdcd1lg1) and *Pdcd1lg2* locus (late stage with f.d.r. = 0.081) as is their receptor *Pdcd1* (late stage). *Pdcd1lg2* is amplified and rearranged frequently in primary

mediastinal large B cell lymphoma[38] and increased expression is speculated to inhibit anti-lymphoma T cell responses[39].

Late-stage biased insertions are located near the H2-D/H2-Q locus (orthologous to the MHC I HLA-B/C loci) and BCL2 transgenic biased clusters are found near the MHC Class II beta chains H2-Ob, H2-Ab1, H2-Eb1, and alpha chains H2-Aa, H2-Ea-ps(orthologous to the MHCII HLA-DRB/HLA-DQB loci) (Supplementary Fig. 11q). Aside from the abovementioned GWAS associations both regions are deleted in human DLBCL[40–42]. There are also suggestive clusters of late-stage insertions surrounding the MHC Class I components H2-T24/T23/T9/T22/BI/T10/T3/Gm7030 (equivalent to the HLA-E locus) and the MHCII alpha/beta chains H2-Oa/H2-DMa/H2-DMb (equivalent to the HLA-DOA/HLA-DMA/HLA-DMB region), suggesting roles for both classical and non-classical MHC components in MuLV lymphoma progression. While loss of MHC loci (or lowered expression) in human B malignancies likely results from selection to escape antitumor immune responses, viruses that drive human lymphoid malignancies (such as Epstein Barr virus and human T cell leukemia virus) encode mechanisms to avoid host immune responses. As such in the context of MuLV infection mutations at MHC loci may represent selection against host antiviral responses rather antitumor responses.

**Negative selection of mutations throughout lymphomagenesis**. Intriguingly we observed many loci throughout the genome that are early-stage specific i.e. undergoing negative selection between early- and late-stage cohorts, suggesting these integrations become detrimental to survival and expansion of developing lymphoma cells. The most significant of these is the Smyd3 locus where intragenic insertions surrounding exons 6–8 are present to a significantly lesser extent in the late-stage lymphomas samples (Supplementary Fig. 11r). SMYD3 is a methyltransferase that methylates H3K4 and H4K5 and overexpression has been observed in a variety of tumor types. SMYD3 methylation of MAP3K2 activates MAP kinase signaling and loss of Smyd3 delays development of both pancreatic and lung tumors[43]. Presumably integrations that disrupt Smyd3 expression are detrimental and hence selected against in lymphomagenesis. SMYD3 loss potentiates the effects of MEK1/2 inhibition on tumor growth[43] and SMYD3 inhibitors have been shown to inhibit the growth of tumor cell lines[44]. The example of the Smyd3 locus demonstrates the potential for time course mutation analysis to not only identify cancer drivers, but also potential targets that while not mutated, are essential for tumor cell growth.

**Co-mutation analyses using subclonal mutations**. Understanding which genes cooperate in lymphomagenesis can inform the biology and subtype of disease; however, co-mutation analyses of subclonal mutations is complicated by the potential presence of multiple independent subclones. When using contingency table tests (such as Fisher's exact test) loci undergoing frequent mutation at the subclonal level will automatically be found co-mutated in the majority of samples (e.g. Gfi1 and Myc). To counter this effect, we performed co-mutation analysis limiting analysis to clonal mutations (NC >0.1). Using only clonal mutations only a handful of loci survived multiple testing correction, (Supplementary Fig. 13a, Supplementary Data 11). To counter the reduced power caused by limiting analysis to clonal mutations, we categorized tumors by clonal mutations at each locus (i.e. classifying each tumor as having an insert in each window or not), and then counted the frequencies of mutations in all other windows (both clonal and subclonal). The latter approach provides greater statistical power than exclusive use of clonal mutations (Supplementary Fig. 13b) but reduces the number of false positives created by loci frequently found to have

subclonal mutations. Importantly this approach incorporating subclonal mutations allows dozens of window pairs to survive multiple testing correction.

We have previously demonstrated that co-mutation analysis can be compromised by the pooled analysis of phenotypically and genotypically distinct groups, which creates false positives from genes that are co-mutated or mutually exclusive due to a primary association with tumor subtype rather than other mutations[6]. For this reason, we also calculated f.d.r. values for each of the genotypes separately (Supplementary Data 11) and devised an online tool that allows subsets of tumors with restricted phenotypes, genotypes, and mutation profiles to be queried. http://mulvdb.org/coocc/index.php.

## Discussion

In this study we present the most comprehensive analysis of MuLV-driven lymphomagenesis produced to date, identifying 700,000 mutations from 521 infected animals with an average of more than 1000 subclonal mutations per sample. By developing a framework that incorporates mutation frequencies of all integrations in both early- and late-stage tumors, we enhance the statistical power enabling the identification of known driving events in cancer and further implicate dozens of novel loci in the biology of lymphomagenesis, in some cases independently of evidence of clonal expansion.

The resolution of mutation coverage illustrates considerable complexity in the position and orientation of selected integrations in the vicinity of verified cancer drivers and suggests uncharacterized locus-specific mechanisms by which these mutations modify expression in a position dependent manner. The online repository (http://mulvdb.org) allows researchers studying lymphoid malignancies to query custom subsets of data for genome-wide associations of a gene of interest with tumor type and mutation status, and create custom tracks for the UCSC genome browser[45]. Tracks on the UCSC genome browser can also be browsed to examine the selection biases at specific loci of interest http://mulvdb.org/coocc/index.php) and subsets of tumors can be queried to identify co-mutated genes within phenotypically/genotypically matched lymphomas.

Recent whole-genome sequencing of cohorts of hundreds of patient samples illustrates the challenges of identifying driver mutations outside the exome[46,47]. Recurrent clonal mutations follow a power law distribution, with statistically intractable rare events making up the bulk of mutations in many tumor types. Proving which of these contribute to disease remains a bottleneck that can only be partly alleviated by larger cohort sizes. While the clonal mutations in MuLV-driven tumors match a similar power law distribution, selection of mutations identified at the subclonal level strongly correlate with clonal mutations and/or known cancer drivers, suggesting that these mutations can provide statistical support for the role of rarely recurrent clonal mutations. Reanalysis of existing cohorts of tumors from other insertional mutagenesis screens alongside equivalent non-malignant tissue may greatly expand the yield of cancer drivers identified as well as eliminate false positives.

Many tumor types display background mutation rates that vary throughout the genome. For instance, mature B cell lymphomas are in part driven by aberrant somatic hypermutation[48]. This variation can confound the identification of driver mutations outside non-coding regions. The overlap we find between independent criteria as evidence of selection (disease stage, tumor type, genetic interactions, and strand bias) using a mutagen that exhibits strong regional variation in distribution demonstrates it is possible to use these criteria as a mitigant of regional variation in mutation frequencies. Furthermore, visualizing this selection as

a continuum at multiple scales using multiple parameters allows intuitive differentiation of recurrent selection in non-exonic regions from mutation hotspots.

Proving that deregulated but intact open reading frames are cancer drivers is problematic, particularly for tumor types such as CLL where the number of coding mutations per cancer genome is low and substantial epigenetic deregulation has been observed[49]. Aside from genes with a supporting role in the literature, this study implicates hundreds of other candidates with equal significance that can be used as a resource to prioritize the study of human cancer drivers and potential therapeutic targets.

Targeted resequencing of recurrently mutated genes in CLL has demonstrated that coding subclonal mutations also undergo significant selection and even convergent evolution[50]. Currently selection of subclonal mutations in cancer is difficult to prove outside the coding regions of known cancer drivers, in part because the error rates of existing NGS platforms limit detection of single-nucleotide allele frequencies to >1%. Novel technologies for detection of lower abundance mutations are in development[51–53], although their throughput and coverage is limited. This study demonstrates the value of applying genome wide, subclonal mutation detection in both malignant and non-malignant tissue. Incorporating these into a framework that combines tumor genotype and phenotype not only provides supporting evidence for rarely mutated cancer drivers but also potentially widens the spectrum of genes encoding therapeutic targets.

## Methods

**Animal work**. All procedures were performed in accordance with the UK Home Office Animals (Scientific Procedures) Act 1986. BCL2-22 (B6.Cg-Tg(BCL2) 22Wehi/J, http://jaxmice.jax.org/strain/002318.html) were bred with wild-type C57BL/6 and BALB/c mice (Charles River, UK). C57BL/6 Vav-BCL2 mice were bred with wild-type BALB/c mice to produce (BALB/c x C57BL/6) F1 Vav-BCL 2 mice.

MuLV was prepared by transfection of 293T cells with the plasmid pNCA[54] (provided by Stephen Goff, Addgene 17363). Newborns were injected intraperitoneally with 50 μl MuLV supernatant. Mice were weighed weekly and monitored three times per week for signs of illness. Mice (infected and matched controls) in the time course cohort were sacrificed and lymphoid organs harvested at predetermined time points, prior to disease onset (9, 14, 28, 42, 56, 84, and 112 days). Survival cohort mice were sacrificed upon developing advanced symptoms of lymphoma and lymphoid organs were harvested and snap frozen in liquid nitrogen immediately. Cell suspensions of spleen tissue were prepared in all cases using the gentleMACS Dissociator (Miltenyi Biotec) set to programme m_spleen_1.01.

**Flow cytometry**. Cryopreserved spleen suspensions were defrosted and washed twice in buffer, PBS-2% fetal calf serum and incubated with 2.0 μg Fc block per $10^6$ cells for 15 min. The samples were then incubated with the antibody cocktails (1 in 200 dilution per antibody) for 15 min after which they were washed. The majority of samples were processed using the Attune NxT Acoustic Focusing Cytometer, Life Technologies, and the remainder were processed on a BD LSRII. All analyses were performed using FlowJo v10.2.

Antibodies used for the general staining panel: Cd3-AF700 Biolegend 197780, Cd4-PerCP-Cy5.5 Biolegend 100434, Cd5-PE Biolegend 100608, Cd8a FITC Biolegend 100706, Cd19-BV421 Biolegend 115538. Additional antibodies: B220/ CD45R-BV510 Biolegend 103247, IgD-APC-Cy7 Biolegend 405716, IgM PE Biolegend 406508, IgG1-FITC BD Bioscience 553443, IgG2a/2b FITC BD Bioscience 553399, IgG3-FITC BD Bioscience 553403, Kappa-AF700 Biolegend 409508, Lambda-APC Biolegend 407306, CD95-PECy7 BD Bioscience 557653, PNA-FITC Vector Laboratories FL-1071. Fc block TruStain fcX™ anti-mouse CD16/32, Biolegend 101320.

**MuLV quantification**. qPCR was performed using primers that had been optimized to exclude amplification of endogenous retroviral sequences similar to MuLV. 5′-GTATGGGCAACTTCTGGCAAC-3′ (forward) and 5′-GAGGGAGG TTAAAGGTTCTTCG-3′ (reverse) amplified a 204 bp region of MuLV in infected mice. Gapdh was used as a control gene using primers 5′-TGCACCACCAACT GCTTAG-3′ (forwards) and 5′-GGATGCAGGGATGATGTTC-3′ (reverse).

For qPCR of MuLV integration copy number DNA concentration was adjusted to 50 ng/μl and 1 μl put into a reaction volume of 20 μl which also included 10 μl of reaction buffer, 0.6 μl of each primer (10 μM stock concentration), and 7.8 μl of $H_2O$. For RTqPCR of MuLV expression levels, RNA was treated with DNase1, (Life

Technologies; 18068-015) using a 0.5 μg input of RNA. cDNA was made using SuperScript II Reverse Transcriptase (Life Technologies; 18064) of random primers and treated with Ribonuclease H (Life Technologies; 18021-071). cDNA was diluted 1/5 and then 1 μl amplified. The MESA Blue qPCR MasterMix Plus for SYBR® Assay No Rox kit (Eurogentec; RT- SY2X-03+NRWOUB) was used for amplification. Cycling conditions were 95 ℃ for 5 min, followed by 39 cycles of 95 ° C for 15 s, 60 ℃ for 60 s followed by a plate read, and then a melt curve from 65 ℃ to 95 ℃ incrementing of 0.5 ℃ every 5 s.

**Integration site cloning and GKC CIS identification**. To clone virus integrations we integrated the method of Koudijs et al.[10] and Uren et al.[11] and modified these for the Illumina platform. DNA was extracted using either the AllPrep DNA/RNA 96 Kit (Qiagen; 80311) or the single sample version of AllPrep DNA/RNA Mini Kit (Qiagen; 80204) as per the manufacturer's instructions. Disposable pestles were used to disrupt tissues in a microfuge tube and the QIAshredder (Qiagen; 79656) was used to homogenize tissues. DNA was quantified using the Qubit® dsDNA HS Assay Kit (Life Technologies; Q32854).

Fifty-five microliters of 20 ng/μl DNA was sheared in a Covaris 96 microTUBE™ Plate (LGC Genomics; 520078) on the Covaris E220 Sonicator with the E220 Intensifier (pn500141). Peak Incident Power 175 W, Duty Factor 10%, Cycles per Burst 200, Treatment Time 55 s. DNA fragments were blunted using NEBNext® End Repair Module (NEB; E6050L) as per the manufacturer's instructions but with a reaction volume modified for the volume of DNA. Blunted fragments were then cleaned using Agencourt AMPure XP magnetic beads (Beckman Coulter; A63880) and A-tailed using the NEBNext® dA-Tailing Module (NEB; E6053L). Samples were again cleaned and ligated to a unique adaptor using T4 DNA Ligase (NEB; M0202L). Ligations were digested with EcoRV-HF® (NEB; R3195L) at 37 ˚C, overnight. Samples were then cleaned and size selected using Agencourt AMPure XP magnetic beads.

Fragments of mouse genome containing MuLV integrations were enriched by nested PCR performed using the Phusion Hot Start II High-Fidelity DNA Polymerase kit (Thermo Scientific, F549L). In the primary PCR a primer to the virus LTR (5′-GCGTTACTTAAGCTAGCTTGCCAAACCTAC-3′) and to the index containing adaptor (5′-AATGATACGGCGACCACCGAGATCTACAC-3′) were used in a 50 μl reaction volume containing 28.5 μl DNA, 10 μl of 5× buffer, 1 μl of 10 mM dNTPs, 2.5 μl of each primer (10 μM), 0.5 μl of Phusion Hot Start II High-Fidelity DNA Polymerase and 5 μl (0.1× final concentration) of SYBR® Green I nucleic acid gel stain, 10,000× in DMSO (Sigma-Aldrich; S9430). qPCR cycling conditions were 98 ℃ for 30 s, followed by 11 cycles of 98 ℃ for 10 s, 66 ℃ for 30 s, and 72 ℃ for 30 s, and finally 72 ℃ for 5 min. Cleaned primary PCR products were quantified using Qubit® and 50 ng was further enriched by using the same adaptor primer and a second nested LTR primer. This primer also contained a second index in order to create more unique index combinations enabling more samples to be pooled per sequencing run. Reaction volumes and cycling parameters were the same as for primary PCR. Secondary PCR products were cleaned and then size selected as described above with a final elution volume of 30 μl.

Each of the 96 samples in each plate were quantified using Qubit® and 25 ng of each sample used to compile a library. Each sublibrary of 96 samples was quantified for amplifiable fragments by qPCR using the KAPA Illumina SYBR Universal Lib Q. Kit (Anachem; KK4824) as per the manufacturer's instructions with DNA dilutions of 1/100, 1/1000, 1/10,000. Equal amounts of each library of 96 were combined to give the final library used for sequencing.

**HiSeq sequencing and insertion mapping**. Libraries were denatured and sequenced on the Illumina HiSeq 2500 using paired end 100 bp reads and dual-indexes of 8 bp following Illumina's standard protocol. Illumina sequencing Bcl files were demultiplexed (based on the barcode index sequences) and FASTQ files generated using Illumina CASAVA version 0.3.0.0. Adapter and LTR sequences were trimmed (allowing two mismatches) and low-quality reads removed. Reads were concatenated in a unique file for each sample and aligned to the mm10 reference mouse genome assembly. For each alignment against the reference, the genomic position and orientation was extracted. The list of paired and mapped reads for each sample was tabulated.

Reads with the same orientation and an LTR position within a window of 10 bases of each other were clustered and considered as a single insertion. All the inserts were built by clustered LTR positions, and each one of them assigned a unique ID, chromosome, orientation, sample ID, minimum LTR position, and maximum LTR position from the alignment. Control uninfected mouse DNA and human DNAs were sequenced to identify any PCR artifacts that might be attributable to cross-contamination of reagents/adjacent wells (which would be found in human DNA) or mispriming events on the mouse genome. All inserts shared between two or more mice were flagged for deletion and only the earliest cloned example of each replicate insertion was kept. Where two replicate inserts were cloned simultaneously one was kept only if it was 10 fold more clonal than all other inserts on the same day. This filtering minimizes the effect of recurrent PCR artefacts and prevents cross contamination from creating false co-mutations. Filtered inserts were used for all analyses except the RNA seq/insert comparison which used unfiltered insertions. All analyses in this paper were performed on a single sample of each spleen. Clonality (relative abundance of individual insertions in a given sample) was then calculated as the number of fragments divided by the

total number of fragments in the sample. NC was calculated by normalizing all insertions in each sample/DNA sample such that the most clonal integration had a value of 1.

**Insertion site statistical analyses**. Identification of CIS by Gaussian Kernel Convolution: CIS loci were identified using the implementation of CIMPL(15) distributed within the KCRBM (16) package (provided by J. de Ridder and J. de Jong). Target genes were automatically assigned using the KCRBM package. Additionally, each locus was manually inspected to assign curated genes.

Co-mutation analyses: Co-mutation analyses were performed using all late-stage inserts within a 100 kb windows surrounding the list of loci identified by GKC. For each locus two-tailed Fisher's exact tests were performed using only clonal inserts (NC>0.1) or by classifying samples with clonal inserts at one locus and then counting the distributions of all inserts at the second locus regardless of clonality.

**Entropy quantitation**. Clonality values among the early-stage samples are relatively uniform, while late-stage samples present few integrations with very high clonality values and most with low clonality values. To quantify this difference and thus be able to order samples from pre-malignancy to late-stage lymphoma we used Shannon entropy[55]. The 50 highest clonality values $c_1, c_2, \ldots, c_{50}$ were transformed into probabilities $p_i$:

$$p_i = \frac{c_i}{\sum_{j=1}^{50} c_j}.$$

The Shannon entropy $E$ over a set of probabilities $p_1, p_2, \ldots, p_n$ is defined as

$$E = -\sum_i p_i \log p_i.$$

The entropy quantifies the spread of a distribution: it is zero when a single $p_i$ is equal to one and all others are equal to zero, and reaches its maximum value when the probabilities are uniformly distributed ($p_i = 1/50$ for every $i$). Probabilities from early-stage samples are closer to a uniform distribution and therefore the samples will have high entropy values, while the probabilities from late-stage samples are closer to a spike, providing low entropy values.

**Clustering**. Samples were clustered based on the shape of clonality profiles. The top 50 ranked clonal insert NC values of each sample were compared to all other samples using dynamic time warp (dtw package) or the Kolmorogov–Smirnov test (ks.dist() function) as a measure of difference between samples (with the R package, function dist() takes as input the list of NC values and the method chosen -"DTW"-). A distance matrix that was clustered using the "average" linkage method (using the R hclust() function).

**Fusion transcript detection**. The Virusfinder package[56] was used to identify high-quality evidence for fusion transcripts between cellular RNA and MuLV transcripts. Chimeric reads mapping to these transcripts were then remapped and visualized using Geneious.

**Statistical analysis**. Genome-wide scanning for selected insertions: A scanning 100kb window is moved across the genome in increments of 10 kb. For each window the number of insertions in each class (early/late, forward strand/reverse strand, BCL2 transgenic/wild type) is counted and the likelihood of this distribution between groups is estimated using two-tailed Fisher's exact test. By comparing neighboring windows, $p$-value minima are identified (i.e. windows where the $p$-value is higher on either side). Where minima are less than 100,000 bp from each other the position with the lowest $p$-value is kept and others discarded. To estimate false discovery rates all insert/group assignments are randomized (e.g. the same number of inserts are early/late but the assignment is random). Local $p$-values are calculated and $p$-value minima are identified. In all, 1000 permutations are used to determine the rate at which each $p$-value is identified by chance.

Cohort comparisons: Survival comparison of cohorts was performed using Prism 6. Significance of the differences in the proportion of B cells in cohorts was determined using Prism 6 Student's $t$-test.

**Ethics approval**. Animal work in this manuscript was carried out in compliance with the Animals (Scientfic Procedures) Act 1996 UK under project license number 70/7353 at designated establishment 70/2722 X32FDCFC1 Imperial College London and experiments were approved by the Imperial College Animal Welfare and Ethical Review Body.

**Code availability**. Custom scripts utilized in this manuscript are available at https://github.com/anthonyuren/MuLV_pipeline for integration site processing, and for any other analyses from the authors upon request.

**Data availability**. The integration site and RNAseq sequence reads mapped to mouse build mm10 are available in the NCBI sequence read archive https://www.ncbi.nlm.nih.gov/bioproject/PRJNA381700 https://www.ncbi.nlm.nih.gov/Traces/study/?acc=SRP110741.

The entire set of insertion sites post mapping can be downloaded from http://mulvdb.org/peaks/tables.php.

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

## Acknowledgements

Charles Bangham, Suzanne Turner, Jesus Gil, Vincenzo de Paulo, Niall Dillon, Matthias Merkenschlager, and Amanda Fisher provided helpful comments on the data and manuscript. Andreas Villunger provided Vav-BCL2 transgenic animals. Emma Mustafa, Aleksandra Czerniak, and Katie Horan assisted with animal care and monitoring. Andre Brown suggested Dynamic Time Warp as a measure of clonality profile similarity. The study was supported by MRC core grant MC_A652_5PZ20. P.W. supported by a Chain-Florey Clinical Fellowship supported by the National Institute for Health Research (NIHR) Imperial Biomedical Research Centre (BRC). B.J.B. supported by an MRC Centenary Award. A.P. and J.C. were supported by BBSRC grants BB/K004131/1, BB/F00964X/1 and BB/M025047/1 to A.P., Consejo Nacional de Ciencia y Tecnología Paraguay (CONACyT) grants 14-INV-088 and PINV15-315, and NSF Advances in Bio Informatics grant 1660648.

## Author contributions

Conception and design: P.W. and A.G.U. Development of methodology: P.W., K.T., B.I., J.C., B.J.B., J.K., L.G., J.E., K.N., and A.G.U. Acquisition of data: P.W., J.C.D., H.D., J.K., K.R., and A.G.U. Analysis and interpretation of data: P.W., J.C.D., H.D., B.I., M.D., J.C., B.J.B., J.K., G.D., M.K., A.P., and A.G.U. Writing, review, and/or revision of the manuscript: P.W., J.D., K.T., B.I., J.C., L.G., J.E., A.P., and A.G.U. Technical and informatics support (mouse maintenance, flow cytometry, reagent generation, database construction and maintenance): P.W., J.C.D., H.D., K.T., B.I., B.J.B., J.J.C., J.K., M.D., M.K., L.G., T.A., J.E., G.T., and G.D.

## Additional information

**Competing interests:** The authors declare no competing interests.

