## [Peer Review File · Nature Communications]

Reviewers' comments:

Reviewer #1 (Remarks to the Author):

This manuscript describes a variety of MuLV-induced models of leukemia/lymphoma in which insertional mutagenesis is monitored over a long time period. The data offer a number of interesting results that pose a variety of hypotheses. However, I am not convinced yet by the quantitative claims made by the authors regarding their detection of specific viral insertion sites in complex samples. In addition, many of the analyses seem superficial in nature.

Major comments

1) Figure S4a shows a sample that has insertion events with relatively high clonality scores (only 1 is < 0.6). How does the quantification look for insertion events that are less abundant in the sample? Did the authors have repeated library preps for samples in the dilution series for the 9:1 or 99:1? This would be informative to see here.

2) I cannot read the axis labels for Figure 2c, but I assume that the maximum value on the y-axis is 1.0. Given that, it would appear that NC values of many of the insertion sites from Day 9 and Day 14 samples are >0.5 . I would disagree with the authors that these insertion events are "sub clonal" as they assert. Perhaps this result is being skewed by the the NC calculation. It might be helpful to see a normalized read number from the HiSeq for this panel instead. If the read numbers for the insertion sites from the early time points are one or two orders of magnitude lower than the tumor samples, then the authors may want to reconsider how the data in Figure 2c are depicted.

3) I do not understand Figure 4a as the legend and the graphic are not consistent. The legend states that the values are normalized to 1 while the graphic shows a scale up to 3.0. Does this mean that the relative abundance of insertions at some loci is 3 times greater in early time points than it is in tumor samples? This isn't clear.

4) I do not understand the main point of Figure 4d. The distinction between late-stage clonal and late-stage sub-clonal events depends entirely on the performance of the sequencing method employed to identify the insertion sites. I am not convinced by the data in Figure S4 that the method has sufficient sensitivity to reliably distinguish clonal from sub-clonal events in tumors. Without additional data showing the performance of the method for insertions in the "sub-clonal" range, I think the authors cannot make significant conclusions regarding clonal vs sub-clonal events in tumors. Moreover, I do not think that this distinction needs to even be made for the major conclusions of the manuscript. Instead I think it's an unnecessary distraction. I would suggest that the authors merge the "late clonal" and "late sub-clonal" data sets into a single "late event" data set.

5) What is the significance of the overlap described between the author's data set and the GWAS studied cited? some overlap is expected simply due to chance. Is the observed degree of overlap much greater than expected? A simple Fisher's Exact test, or something comparable, should be reported. If the results are not significant, this section should be removed.

6) The results of the gene ontology analysis need to be paired down. The authors currently report over 5000 GO terms in the supplemental table S8. Over 1000 of these have an FDR < 0.05 . This draws into question the relevancy of the results that are highlighted in the manuscript. With such a large number of hits, how meaningful is it that these specific categories appear on this list? The authors use these results to suggest that classical and non-classical MHC pathways are involved in lymphoma progression. It is unclear why these pathways are emphasized while hundreds more go without mention. Moreover, the authors fail to acknowledge that their leukemia/lymphoma model is virally-induced. Prior work has implicated MHC in the immune response to MuLV. Given this prior

work, what is the evidence that the MHC connection in this model is relevant for human disease that is not virally-induced?

Minor comments

Figure 3e & f

Add labels to y-axis

Figure 4a-c

Add labels to y-axis

Reviewer #2 (Remarks to the Author):

This manuscript describes the use of retroviral mutagenesis modeling to establish lymphoid malignancies in WT or BCL2 transgenic mice (to promote B lineage disease) with state of the art Illumina next generation sequencing to identify retroviral integrations. The authors profile the lineage and differentiation stage of the tumours and use two experimental and computational approaches to identify subclonal events that may then be described as driver lesions. The experimental approach is sacrificing cohorts of mice at early time points (that are deemed "preleukemic") and comparing these with mutations identified at later time points. The second is to use computational approaches - entropy and dynamic time warping - to infer these lesions. The authors spend some time focusing on the type of integration (E.g. strandedness) and specific sites (such as the well described MYC/PVT region).

The authors are experts in the use of these models, and the efforts to extract, model and exploit these subclonal data are impressive. I have relatively few concerns about the execution of the experiments, but I do have several major concerns about the design of the study, and whether it is able to definitively address the notion that subclonal lesions are drivers.

My greatest concern is intra-animal tumor heterogeneity - that is that the majority of animals clearly have at least 2 tumors of different lineage. This suggests that there are multiple genetically independent tumor populations - probably many more - and that if there are subclonal lesions that change over time, they reflect one distinct tumor winning out over time, rather than reflecting human disease, where multiple clones from a founding ancestral clone are propagated, and subclonal genetic lesions can emerge over time. This multi-tumor composition also affects analysis and interpretation. If there are at least two completely distinct tumors (one B, one T for example) they may have "clonal" lesions specific to each lineage of tumor that the current approach may misinterpret as subclonal (drivers). Such lesions can then become predominant if one tumor simply outgrows another. This may lead to the erroneous conclusion that these "subclonal" lesions are drivers - they may have been clonal in that tumor throughout. Sorting the tumor populations into each lineage, and/or clonality analysis, and/or single cell analysis would be required to resolve this.

It is also unclear if the serial samples address the questions posed. These do not reflect the evolution of a tumor over time, but a sampling of a cohort at different time points. The authors also state that these are preleukemic - presumably as the animals are not moribund - but this doesn't mean they are not preleukemic - one would have to purify the populations and perform secondary transplants to address this.

The data presented are limited to presentation of integration sites only. There is no consideration of other modalities of genomic alteration - DNA copy number alterations, sequence variations - that do arise in such screens and may impact the landscape of alterations described, and may also influence clonality analysis (e.g. see Dang et al Blood 2015).

A concern is inference of clonality from the sequencing data. There is apparently no stringency at all - as few as 1 read is called as an integration. The method is also unclear. For variant calls, one would compare mutant v germline allele fraction. Here the normalization appears to be to the sheared fragments - but wouldn't this be genome wide, not a site specific analysis? Many other factors including adapter ligation efficiency, insert size, PCR amplification efficiency, clustering efficiency etc would have an effect on the N of reads per site. It is not clear that the dilution series of a clonal variant addresses this.

If the analysis is to be restricted to RIS analysis, this is not developed enough. The inference is based on gene name and proximity, with no consideration of effect (Activation? inactivation?) of the insertion; RNA_seq of a subset of samples would address this.

The ability to interpret and have confidence in the authors conclusions is limited by the difficulty in understanding what the new algorithms achieve - the Entropy and Dynamic Time Warping analyses.

It would be of interest to have a clearer presentation of new genes (if any) identified by this study. Many (e.g. SFig 5) appear to be well credentialled genes (MYC, NOTCH1, IKZF1 etc) that are shown as Late genes - to this reviewer this seems the authors are suggesting these are subclonal drivers. This does not recapitulate human disease. Extending this, there is no presentation of how faithfully these models represent the genetics of human B-ALL and T-ALL.

As a minor point, it is not clear why human and mouse normal DNA were sequenced.

Reviewer #3 (Remarks to the Author):

Tracking Subclonal Mutation Frequencies Throughout Lymphomagenesis Identifies Cancer Drivers in Mouse Models of Lymphoma

This manuscript describes the use of the murine leukemia virus to catalogue the integration sites genome-wide that demonstrate patterns indicative of driving lymphomagenesis. The authors employ some novel approaches to map integration sites and deconvolute clonal from sub-clonal selection and several transgenic backgrounds (Vav-BCL2, Emu-BCL2, wild-type) to aid in promoting B-cell lymphomas. The authors characterized ~500 induced B-cell lymphomas and T-cell ALLs using their integration site sequencing strategy that allowed detailed characterization of the evolutionary/selection history in each mouse. The integrations are examined with respect to strand bias, which can indicate a preference towards events that promote expression of nearby genes.

In general, this work promotes an increased statistical power in their somewhat novel design that was achieved by leveraging the signature of selection for subclonal as well as clonally-selected integrations among the premalignant and malignant cells, respectively. This is framed as a key feature of the study that sets it apart from previous attempts to map cancer drivers using viral integration screens. Their strategy is very clearly supported by the recapitulation of an extensive set of genes orthologous to known human cancer genes including many relevant to B-cell lymphomas. This is an interesting and novel study that yields a potentially invaluable resource for researchers who study the genetic nature of lymphomas.

Major issues

Comparison to the cancer gene census is central to many of the displays and tables. I am worried the authors failed to ensure whether there are recently described lymphoma-related genes absent from the census. For example, MEF2B, GNA13, S1PR2 are fairly well established lymphoma genes that are lacking from the CGS. There are quite likely others missing as well. It seems reasonable to add a more curated set of genes that are relevant to the diseases being studied here.

I was unfamiliar with the dynamic time warping methodology and, although the application seems valid, it appears to (potentially) be an unnecessary complication to the paper. In Figure 3, it seems that the Shannon entropy distribution plotted as a histogram would have an obvious cut point around 3.5. This is only shown as boxplots (panel C) after the two groups were defined and the distribution (by rank), in panel B so it is not possible to tell from the data whether the approach used to make panel A was actually needed. Further, even if the entropy score alone does not capture enough of the information from each mouse, it seems that a Spearman correlation would be applicable here since the integrations are being described by rank already.

P values are used in quite a few of the tables and display items but the statistical approaches used to derive them are not explained in sufficient detail. Here are some examples:

Contingency table tests used for determining significant integration sites. A formal explanation of how these are performed using these data should be included in the Methods. The same goes for the contingency tests used to generate Figure S7. Furthermore, were the P values corrected for the many tests that were performed? If so, how?

Supplemental Figure S6 is a very useful resource as I'm sure the browser will be. There is an unsatisfying paucity of integration with any genomic annotations beyond genes and the cancer census. These figures indeed raise some interesting trends that are worth some discussion in the text. PRDM1, for example, has recurrent integrations spanning an extremely large region. It is difficult to discern how these can all be impacting PRDM1 expression. The REL locus is also quite interesting because there appears to be two regions that are enriched for integrations (upstream of REL proper and downstream of BCL11A). This is suggestive of distal regulatory elements that may be enhanced by these integrations. Are there existing data sets for mouse that would reveal whether there is coincidence of integration sites and potential regions that act as enhancers in mature B cells? If such data exist, it would be worthwhile to discuss this in the context of the present data.

Supplemental Figure S4 - The "most clonal" integrations are shown (11 of them, it seems). How were these selected? This does not appear to be the top 50, which is what was used for many of the other displays. Ideally there should be some indication of the cutoff and/or another plot showing the lower clonality sites for this replicate.

Minor issues

Strand bias - when referring to a bias for the forward strand, is this in reference to the strand of the gene to which the event was assigned or the strand of the chromosome? I am fairly certain this must be the genome because many events list more than one gene but this should be clarified or perhaps the strand of the gene should be noted in figures because strand bias is probably more valuable when considered in the context of the nearby gene.

Figure S3 could be improved for clarity. The last few steps of the sequencing (i.e. reading the barcodes) are somewhat irrelevant and don't add anything the figure unless I have missed something. It would be better if the focus of this figure was on how the method differs from the published procedure from which it was derived.

Figure S7 - why are these not symmetrical along the diagonal?

This analysis could be quite informative but the result is not readily interpretable as presented. Without knowing if/how P values were corrected it is unclear how much of what is seen as green is worth considering. A table ranked by pairs of loci might be superior.

The heat maps that are largely black with green, blue etc in the foreground could benefit from a different colour scheme. For example, white background and a Brewer palette representing the gradient of P values spanning significance levels.

Spelling error in legend for Table S7 - lymphoma

Why was this comparison only done using published copy number data? Since many of the integrations in this study were mapped to specific genes, it seems that the genome and exome studies would also be worth comparing to. There are several such studies. They could also be used to obtain a superior list of genes beyond those in the census for some of the comparisons discussed above.

'Tracking Subclonal Mutation Frequencies Throughout Lymphomagenesis Identifies Cancer Drivers in Mouse Models of Lymphoma.' Webster *et al.*

Reviewers' comments:

Reviewer #1 (Remarks to the Author):

This manuscript describes a variety of MuLV-induced models of leukemia/lymphoma in which insertional mutagenesis is monitored over a long-time period. The data offer a number of interesting results that pose a variety of hypotheses. However, I am not convinced yet by the quantitative claims made by the authors regarding their detection of specific viral insertion sites in complex samples. In addition, many of the analyses seem superficial in nature.

We appreciate the concerns regarding our quantitative claims and have now tried to better explain the rationale for and methodology of insert quantitation. We have included additional data and analyses to verify the reproducibility of our quantitation. We have also addressed several oversights in the comparative analyses of our data with human cancer genomes, which were pointed out by the reviewer, and we included new comparisons to human tumor genomes and RNAseq analysis of a subset of tumors.

Major comments

1) Figure S4a shows a sample that has insertion events with relatively high clonality scores (only 1 is < 0.6). How does the quantification look for insertion events that are less abundant in the sample?

The raw and normalized clonality values of inserts are reproducible only for clonal inserts and as expected, become noisier when inserts are represented by a single DNA fragment/read/cell. To explore this further we have analysed data from 2 series of 6 replicate libraries taken from 2 spleen DNA samples. The results from these replicate libraries are now included as supplementary figures S4a-e. These replicate series indicate that inserts that have a normalized clonality of 0.1 or above are found with a high degree of quantitative reproducibility whereas the vast majority of reads at lower clonalities, including the majority of single read integrations are only rarely found in more than one replicate. We have mentioned this analysis in the text (pp.8-9) and added a detailed explanation of this comparison in the legends of supplementary figure S4a-e.

Did the authors have repeated library preps for samples in the dilution series for the 9:1 or 99:1? This would be informative to see here.

We think that the repeat libraries discussed above can address this concern. The dilution experiment included 5 mixed samples where the clonal inserts from both DNAs behave as expected and proportions of the inserts within each sample remain consistent between samples. This demonstrates that read/fragment abundance is proportional to dilution with a slight bias for the lower clonality inserts and against the higher clonality ones.

2) I cannot read the axis labels for Figure 2c, but I assume that the maximum value on the y-axis is 1.0. Given that, it would appear that NC values of many of the insertion sites from Day 9 and Day 14 samples are >0.5. I would disagree with the authors that these insertion events are "sub clonal" as they assert. Perhaps this result is being skewed by the NC calculation. It

might be helpful to see a normalized read number from the HiSeq for this panel instead. If the read numbers for the insertion sites from the early time points are one or two orders of magnitude lower than the tumor samples, then the authors may want to reconsider how the data in Figure 2c are depicted.

It is true that presenting the normalized clonality values without non-normalized values could be easily interpreted as premalignant samples appearing to have clonal insertions. We have added plots without normalization to show that the number of fragments for the most abundant inserts is a much lower fraction of the sample in early samples than in late samples. We've also altered these figures to enlarge the axes markers and explained the y and x axes more clearly in the legend.

3) I do not understand Figure 4a as the legend and the graphic are not consistent. The legend states that the values are normalized to 1 while the graphic shows a scale up to 3.0. Does this mean that the relative abundance of insertions at some loci is 3 times greater in early time points than it is in tumor samples? This isn't clear.

We apologize for the confusion; the reviewer interpreted the data correctly, and we have now clarified this in the legend (this figure is now figure S5). For each window in the time series all the time points were normalized such that the final late stage disease time points are all set to 1, i.e. we are comparing how all top 50 clonal CIS windows progress over the time course. Looking at windows around individual loci, it is clear that some stochastic noise is present over the time course. Certain loci may also be selected against (the example of Smyd3 is discussed on p. 18 in the results/discussion). The global trend is clearer when viewing the median and mean values in Fig S5b) and S5c) but we felt it was important to show the values and stochasticity of individual loci as well.

4) I do not understand the main point of Figure 4d. The distinction between late-stage clonal and late-stage sub-clonal events depends entirely on the performance of the sequencing method employed to identify the insertion sites. I am not convinced by the data in Figure S4 that the method has sufficient sensitivity to reliably distinguish clonal from sub-clonal events in tumors. Without additional data showing the performance of the method for insertions in the "sub-clonal" range, I think the authors cannot make significant conclusions regarding clonal vs sub-clonal events in tumors. Moreover, I do not think that this distinction needs to even be made for the major conclusions of the manuscript. Instead I think it's an unnecessary distraction. I would suggest that the authors merge the "late clonal" and "late sub-clonal" data sets into a single "late event" data set.

At the onset of the study, we could not say it was justified to include all subclonal/single read mutations in the analysis and felt that treating all mutations as equivalent would be discarding (admittedly partial) information about mutation abundance. Our best estimate was that of the of 700,000 mutations in the data set only 3,000 were clonal. We felt it was important to demonstrate as thoroughly as possible whether low clonality integrations should be excluded or if their inclusion would be useful and could be justified. The main point of Figure 4d (now Figure S5d) was to demonstrate that within the top 50 CIS loci, subclonal and clonal events perform similarly. We are not attempting a precise distinction between clonal and subclonal events, but rather would like to show that subclonal events are informative. We have now moved this analysis to the supplementary material to avoid distracting the reader. After having justified the use of all inserts in this manner, the subsequent analyses in the manuscript use pooled late stage data, as suggested by the referee.

5) What is the significance of the overlap described between the author's data set and the GWAS studied cited? some overlap is expected simply due to chance. Is the observed degree of overlap much greater than expected? A simple Fisher's Exact test, or something comparable, should be reported. If the results are not significant, this section should be removed.

To place a p-value on this analysis is less trivial than it seems. This is due to the highly variable and overlapping scope of most lymphoid malignancy GWAS studies (some cohorts have been extended and republished many times and generally they find the same loci). Nonetheless having performed an extensive review and summary of lymphoid malignancy GWAS cohorts, we have taken all disease SNPs identified in a series of 18 studies published since 2009 (supplemental table S10), mapped these SNPs over to the mouse genome and shown significant overlap of our loci in a candidate gene independent manner (using Fisher's exact test of the 26 GWAS loci of combined studies we find 14 correspond to our CIS, the expected number by chance would be 0.82).

6) The results of the gene ontology analysis need to be paired down. The authors currently report over 5000 GO terms in the supplemental table S8. Over 1000 of these have an FDR < 0.05. This draws into question the relevancy of the results that are highlighted in the manuscript. With such a large number of hits, how meaningful is it that these specific categories appear on this list?

We appreciate the reviewer's point and have removed this table in the current version since we don't feel it adds a great deal to the analysis and was a token inclusion for descriptive purposes "Gene ontology analysis of the entire list of candidates yields search terms that are dominated by lymphocyte biology, notably lymphocyte activation, lymphocyte aggregation and cell-cell adhesion" We felt this sentence represented the gist of the top 30 terms in the list.

The authors use these results to suggest that classical and non-classical MHC pathways are involved in lymphoma progression. It is unclear why these pathways are emphasized while hundreds more go without mention. Moreover, the authors fail to acknowledge that their leukemia/lymphoma model is virally-induced. Prior work has implicated MHC in the immune response to MuLV. Given this prior work, what is the evidence that the MHC connection in this model is relevant for human disease that is not virally-induced?

We felt it worthwhile to discuss costimulatory molecules and MHC loci since they were more prevalent in this study than prior insertional mutagenesis studies. We've now included mention of these on p.17, which is arguably a better place to discuss them since these loci tend to be BCL2 and B cell specific and we cite corroborating evidence from human studies and GWAS of non-viral mature B lymphoid malignancies. It is certainly true, as the reviewer pointed out, that some mutations of immune mediator/modulator genes might also be selected as a consequence of virus host interactions and we have now explicitly mentioned this caveat on p.17.

"Whilst loss of MHC loci (or lowered expression) in human B malignancies likely results from selection to escape anti-tumour immune responses, viruses that infect human lymphoid cells (including those that cause malignancies such as Epstein Barr Virus and Human T-cell leukaemia virus) encode mechanisms to avoid host immune responses. In the context of MuLV infection, mutations at MHC loci may therefore also represent selection against host antiviral responses rather than antitumor responses."

Minor comments

Figure 3e & f
Add labels to y-axis

Figure 4a-c
Add labels to y-axis

These figures have been modified as suggested. Figure 4 is now Figure S5.

Reviewer #2 (Remarks to the Author):

This manuscript describes the use of retroviral mutagenesis modeling to establish lymphoid malignancies in WT or BCL2 transgenic mice (to promote B lineage disease) with state of the art illumina next generation sequencing to identify retroviral integrations. The authors profile the lineage and differentiation stage of the tumours and use two experimental and computational approaches to identify subclonal events that may then be described as driver lesions. The experimental approach is sacrificing cohorts of mice at early time points (that are deemed "preleukemic") and comparing these with mutations identified at later time points. The second is to use computational approaches - entropy and dynamic time warping - to infer these lesions. The authors spend some time focusing on the type of integration (E.g. strandedness) and specific sites (such as the well described MYC/PVT region).

The authors are experts in the use of these models, and the efforts to extract, model and exploit these subclonal data are impressive. I have relatively few concerns about the execution of the experiments, but I do have several major concerns about the design of the study, and whether it is able to definitively address the notion that subclonal lesions are drivers.

My greatest concern is intra-animal tumor heterogeneity - that is that the majority of animals clearly have at least 2 tumors of different lineage. This suggests that there are multiple genetically independent tumor populations - probably many more - and that if there are subclonal lesions that change over time, they reflect one distinct tumor winning out over time, rather than reflecting human disease, where multiple clones from a founding ancestral clone are propagated, and subclonal genetic lesions can emerge over time.

We appreciate these concerns and think they might arise at least in part from lack of clarity in how we described our aims and the model used. Our study is not intended as an equivalent to exome resequencing tumor lineage studies from multiple biopsies of human tumors. It is rather an analysis of the entire mutational spectrum of malignant clonal outgrowths in parallel to subclonal and premalignant/non-malignant tissues. As such we have:

- Taken the most clonal mutations as a gold standard for this model.
- Compared this loci list to known human cancer genes.
- Expanded analysis to include subclonal/low abundance mutations (that outnumber clonal mutations by >100 to 1).
- Demonstrated the additional usage of the subclonal identifies additional loci that are also human cancer genes.

There is no real human equivalent to this study because the sensitivity of insertion mutation cloning as has no real equivalents in human cohorts. We demonstrate that low abundance mutation selection is observed at loci that only rarely contribute to malignant clonal outgrowth and as such can be used to strengthen the case for selection of driver mutations

that only rarely achieve clonal outgrowth.

We also now explicitly state the nature of MuLV disease in the text.

“A single clonal outgrowth of pure tumor cells containing few mutations will yield high coverage of each mutation, whereas a clonal outgrowth with dozens of concurrent mutations alongside a large proportion of non-tumor DNA will yield low coverage for even the most clonal mutation. Additionally, there may be multiple independent clones of varying clonality within each tumour and multiple independent subclones of each lymphoma.”

Mouse models of cancer with 100% penetrance must statistically have multiple competing premalignant/malignant clones at different stages of development. In the human scenario, the extent to which driver mutations are also represented in premalignant clones/non-tumor is unknown for all but a few genes, but we agree they are unlikely to be as abundant as seen in our model. Nevertheless, we feel the comparison and joint analysis of all mutations from both malignant and premalignant cells can greatly increase the confidence of genes identified as drivers of clonal selection. Although we can't distinguish between the subclonal mutations present in the dominant tumor clone vs single bystander lymphocytes, we can say that aggregate analysis of both is informative.

This multi-tumor composition also affects analysis and interpretation. If there are at least two completely distinct tumors (one B, one T for example) they may have "clonal" lesions specific to each lineage of tumor that the current approach may misinterpret as subclonal (drivers). Such lesions can then become predominant if one tumor simply outgrows another. This may lead to the erroneous conclusion that these "subclonal" lesions are drivers - they may have been clonal in that tumor throughout. Sorting the tumor populations into each lineage, and/or clonality analysis, and/or single cell analysis would be required to resolve this.

We would like to clarify that the vast majority of our mutations are indeed not driver mutations. We merely demonstrate that in small scale biopsies (averaging 1-2mm in size) there are enough subclonal events corresponding to known driver mutations that the data can be used to supplement analysis of clonal mutations i.e. the findings of both datasets overlap sufficiently to justify the use of both. We have tried to avoid stating that these mutations are “subclonal drivers” but rather would say that subclonal mutations are indicative of driver status often enough to be useful data. We don't feel this is a given.

Regarding separation of lineages or clones, sorting by immunophenotype doesn't allow separation of independent clones within lineages. For a small number of cases we have separated B & T cell populations prior to insert cloning but found that the insert cloning method is so sensitive that even “pure” populations contain a fraction of the inserts of the other lineage. This is an inherent caveat of highly sensitive PCR based methodologies. We do observe percentages of B & T lineages and mutation frequencies correlate with these, but we have avoided overinterpreting these associations.

It is also unclear if the serial samples address the questions posed. These do not reflect the evolution of a tumor over time, but a sampling of a cohort at different time points. The authors also state that these are preleukemic - presumably as the animals are not moribund - but this doesn't mean they are not preleukemic - one would have to purify the populations and perform secondary transplants to address this.

The time course experiment is used to compare mutation frequencies from the early/late time points; it is not an attempt at lineage tracing within individual animals. The early/late

comparision successfully identifies known cancer driver loci and demonstrates that strand bias increases at later points in tumor development, thus further validating our approach. We cannot make claims about mutation order within each sample, but simply observe abundance of mutations at different loci in populations of different ages.

For want of a better term, we had referred to premalignant tissue/samples/mutations, but whenever possible, we have now changed these terms to “early stage” or “late stage” tissue and mutations. It is correct that we do not know if all biopsies taken from “pre-malignant” animals contain cells that will eventually become malignant, we can only say that the disease is 100% penetrant.

The data presented are limited to presentation of integration sites only. There is no consideration of other modalities of genomic alteration - DNA copy number alterations, sequence variations - that do arise in such screens and may impact the lanscape of alterations described, and may also influence clonality analysis (e.g. see Dang et al Blood 2015).

The study mentioned (Dang et al. <https://doi.org/10.1182/blood-2015-02-626127>) is one of the better examples of why we decided against multimodal analyses. This study finds an average of 25 nonsynonymous coding mutations per MuLV tumour (n=10) which is fivefold less than found in their ENU cohort. Within the MuLV tumours, only 15 genes were recurrently mutated. The CGH of this study of the 3 MuLV tumors analyzed only one copy number change (trisomy 15) was observed in one sample. In the entire study, copy number aberrations were sparse and generally represented events that had been previously described e.g. loss of *Ikzf1*, *Nf1*, *Cdkn2a/b*.

Other studies with MuLV and other insertional mutagens have found little or no copy number changes and very little exonic sequence variation primarily because these models are driven by deregulation of ORFs (equivalent to translocations and/or non-coding mutations) and occasionally intragenic mutations (which can be truncating/inactivating/stabilizing). We could have included similar analyses, but the yield of new findings would arguably be minimal compared to the cost. We find that more benefit is gained from comparisons to human tumor datasets.

A concern is inference of clonality from the sequencing data. There is apparently no stringency at all - as few as 1 read is called as an integration. The method is also unclear. For variant calls, one would compare mutant v germline allele fraction.

We have better illustrated the method in a revised Figure S3. Our insert mutations are identified based on two nested rounds of ligation mediated PCR and a further nested sequencing primer. We are only amplifying mutant DNA and are therefore unable to assess the mutant/germline fraction. The method is a tradeoff between sensitivity (nearly everything sequenced represents a mutation) vs quantitation. The replicate samples demonstrate that the method is capable of reproducibly finding the most clonal insertions and these are also identified by whole genome sequencing. Although single reads of SNVs cannot be distinguished from sequencing errors, single reads of translocations/insertions/deletions can be readily distinguished from sequencing error and the only grounds for excluding them is if they are artefacts of library construction.

We have also resequenced some of our libraries using two primers that are set further back from the virus genome junction and have found that between 95-97% of read pairs with read 2 phred score cutoff of >30 correspond to a virus genome junction. Low quality read 2 presumably stems from an inability of the read 2 primer to bind the library due to lack of an

LTR sequence i.e. a PCR product that doesn't contain the end of the LTR cannot be sequenced. The small number of high quality read 2 that don't contain a virus genome junction appear to correspond to ligation and PCR artifacts that either fail to map to the mouse genome or are removed by filtering steps during final processing of the data.

It is ambitious to use all inserts including those represented by a single read, and this is why we went to such lengths to justify the use of all mutations, but readers who would like to use the dataset for subsequent analyses are able to filter according to their preferred read depth or clonality.

As a minor point, it is not clear why human and mouse normal DNA were sequenced.

The uninfected mouse DNA and human DNA is sequenced to identify any PCR artifacts that might be attributable to cross contamination of reagents/adjacent wells (which would be found in human DNA) or mispriming events on the mouse genome (there are about 18 endogenous retroviral loci that loosely match our primer sets within the mouse germline). We now explain the use of control DNAs in the supplemental methods section pp. 4-5.

Here the normalization appears to be to the sheared fragments - but wouldn't this be genome wide, not a site-specific analysis? Many other factors including adapter ligation efficiency, insert size, PCR amplification efficiency, clustering efficiency etc. would have an effect on the N of reads per site. It is not clear that the dilution series of a clonal variant addresses this.

We don't wish to oversell the quantitation other than to say it is reproducible (now better illustrated in supplemental Fig. S4) and demonstrably concentration dependent in a dilution series i.e. the same mutation will be sequenced with roughly 100-fold less coverage if it is 100-fold less abundant. This does not eliminate differences between loci and there will certainly be blind spots in the genome where some inserts will be underrepresented or missed but this is true of any genome wide study. We have added a paragraph to the supplemental methods section (pp. 4-5) that outlines the greatest potential sources of error in our protocol.

"Several potential sources of error may influence quantitation based on fragment counts. The EcoRV digestion step (designed to remove sequences that run into the virus from the 3' LTR) means approximately 5% of the genome within 150 bases of an EcoRV site will be under represented or eliminated. Sequence dependent amplification bias (e.g. GC rich regions being underrepresented) is minimized by cutting the number of PCR cycles down from a total of over 50 in prior protocols to 25. Furthermore, some inserts corresponding to highly recurrent sites (such as the 3' UTR of *Mycn*) may be eliminated in the final filtering stage, because they can be mistaken for recurrent PCR artifacts and/or cross contamination."

If the analysis is to be restricted to RIS analysis, this is not developed enough. The inference is based on gene name and proximity, with no consideration of effect (Activation? inactivation?) of the insertion; RNA_seq of a subset of samples would address this.

We employed the automated prediction of targets using the KCRBM software package that was originally trained on a set of MuLV lymphoma expression arrays. We have now included the list of most likely mechanisms of nearby gene deregulation that are generated by this software package in table S2.

As mentioned above we have now also performed RNAseq on 26 samples. Subclonal integrations would not be expected to yield significant alterations in expression. Within this

cohort there are too few samples with clonal integrations at any given locus to demonstrate statistical significance for deregulated expression. Nonetheless by looking for modified transcripts we find a handful of expected fusion transcripts of *Mycn* and *Notch1* (as described in previous screens) and the degree of clonality corresponds with the abundance of these transcripts.

The ability to interpret and have confidence in the authors conclusions is limited by the difficulty in understanding what the new algorithms achieve - the Entropy and Dynamic Time Warping analyses.

We have now explained this more clearly. The entropy approach we used to stage tumors is similar to the Shannon entropy approach used by *Brown et al.* and *Baldow et al.* (references 13 & 14). Nonetheless we wanted to employ two independent methods for staging samples. To this end we employed a novel approach to cluster based on the use of distance measures (specifically using dynamic time warp). DTW it is a well-accepted methodology for comparison of ordered data series. We have also now used the Kolmogorov Smirnov statistic and obtained nearly identical groupings.

It would be of interest to have a clearer presentation of new genes (if any) identified by this study. Many (e.g. SFig 5) appear to be well credentialled genes.

The genes shown were chosen to demonstrate that subclonal events are indicative of the drivers found in human data. The definition of “new” genes is somewhat fuzzy. Many of the genes listed in table S5 are from single studies that require further confirmation. The remaining loci from our study have no prior evidence we could find in the literature or are evidenced in the newly added comparison to human exome studies of lymphoid malignancies (included at the request of reviewer 3).

We have now combined and compared our screen with human copy number, GWAS and exome studies as well as curated lists of cancer genes. Cancer genome datasets invariably generate a few unequivocal drivers and a much longer list of weaker candidate genes, whose functional relevance can only be established by integration with other datasets or follow up studies. Our study generates hundreds of loci that survive correction for multiple testing. More than 100 of these have precedent in the literature and we feel the remainder are a valuable resource for prioritizing the study of other candidates.

Reviewer #3 (Remarks to the Author):

Tracking Subclonal Mutation Frequencies Throughout Lymphomagenesis Identifies Cancer Drivers in Mouse Models of Lymphoma

This manuscript describes the use of the murine leukemia virus to catalogue the integration sites genome-wide that demonstrate patterns indicative of driving lymphomagenesis. The authors employ some novel approaches to map integration sites and deconvolute clonal from sub-clonal selection and several transgenic backgrounds (Vav-BCL2, Emu-BCL2, wild-type) to aid in promoting B-cell lymphomas. The authors characterized ~500 induced B-cell lymphomas and T-cell ALLs using their integration site sequencing strategy that allowed detailed characterization of the evolutionary/selection history in each mouse. The integrations are examined with respect to strand bias, which can indicate a preference towards events that promote expression of nearby genes.

In general, this work promotes an increased statistical power in their somewhat novel design that was achieved by leveraging the signature of selection for subclonal as well as clonally-selected integrations among the premalignant and malignant cells, respectively. This is framed as a key feature of the study that sets it apart from previous attempts to map cancer drivers using viral integration screens. Their strategy is very clearly supported by the recapitulation of an extensive set of genes orthologous to known human cancer genes including many relevant to B-cell lymphomas. This is an interesting and novel study that yields a potentially invaluable resource for researchers who study the genetic nature of lymphomas.

Major issues

Comparison to the cancer gene census is central to many of the displays and tables. I am worried the authors failed to ensure whether there are recently described lymphoma-related genes absent from the census. For example, MEF2B, GNA13, S1PR2 are fairly well established lymphoma genes that are lacking from the CGS. There are quite likely others missing as well. It seems reasonable to add a more curated set of genes that are relevant to the diseases being studied here.

This is a good suggestion and we have now included a new analysis comparing our data to exomes of several lymphoid malignancy cohorts and found significant overlap (Figure 6 pp.13-14): 78 MuLV candidate genes were found mutated in 2 or more samples of these studies.

I was unfamiliar with the dynamic time warping methodology and, although the application seems valid, it appears to (potentially) be an unnecessary complication to the paper. In Figure 3, it seems that the Shannon entropy distribution plotted as a histogram would have an obvious cut point around 3.5. This is only shown as boxplots (panel C) after the two groups were defined and the distribution (by rank), in panel B so it is not possible to tell from the data whether the approach used to make panel A was actually needed. Further, even if the entropy score alone does not capture enough of the information from each mouse, it seems that a Spearman correlation would be applicable here since the integrations are being described by rank already.

We appreciate that the use of DTW in this context is novel, but we included it because it is an entirely independent approach that yields a similar result to the use of entropy scores. While the use of the Spearman correlation is not entirely appropriate here (because it is a rank-based test), based on this suggestion we have now also used the Kolmogorov Smirnov statistic as another distance measure, which gave nearly identical groupings as obtained by DTW.

P values are used in quite a few of the tables and display items but the statistical approaches used to derive them are not explained in sufficient detail. Here are some examples: Contingency table tests used for determining significant integration sites. A formal explanation of how these are performed using these data should be included in the Methods. The same goes for the contingency tests used to generate Figure S7. Furthermore, were the P values corrected for the many tests that were performed? If so, how?

We apologize for not providing sufficient detail and have now tried to make all statistical methodologies more apparent to the reader. The statistics of our early/late, genotype, cell type and strand bias were only described in the legends of supplementary tables, but we have now clarified this in the main text. All contingency table tests were Fisher's exact test and whilst the methodology is explained in the legends for supplementary tables we have now added it to the Methods sections and clarified this in the text. Figure S7 has been

updated to include multiple testing correction in the tabulated version of results. In this analysis dozens of commutated gene pairs survive when analyzing the larger cohorts, which would not be the case were the study limited to analysis of clonal integrations.

Supplemental Figure S6 is a very useful resource as I'm sure the browser will be. There is an unsatisfying paucity of integration with any genomic annotations beyond genes and the cancer census. These figures indeed raise some interesting trends that are worth some discussion in the text. PRDM1, for example, has recurrent integrations spanning an extremely large region. It is difficult to discern how these can all be impacting PRDM1 expression. The REL locus is also quite interesting because there appears to be two regions that are enriched for integrations (upstream of REL proper and downstream of BCL11A). This is suggestive of distal regulatory elements that may be enhanced by these integrations. Are there existing data sets for mouse that would reveal whether there is coincidence of integration sites and potential regions that act as enhancers in mature B cells? If such data exist, it would be worthwhile to discuss this in the context of the present data.

Mapping MuLV integrations with respect to genomic features such as promoters, expressed genes and enhancers has an extensive literature in its own right and this is why we have gone to such lengths to differentiate between unselected and selected integrations by multiple criteria without sole reliance on density. The potential for retrovirus integrations to act over large distances has been documented including a very thorough recent study by Babaei *et al.* (<https://www.nature.com/articles/ncomms7381> ref 28). Whether allocation of candidate genes is automated or curated will impact the results, but this is also true of cancer studies of GWAS, epigenomes and copy number variation. We have now elaborated on these issues in the results (p. 15).

“MuLV is known to deregulate genes via a diverse set of mechanisms including distal interactions of virus enhancers via 3D conformation of the genome²⁸. The integration bias of MuLV for enhancers²⁹ and capacity for long range interactions likely explains the clusters of late stage/strand biased/genotype specific integrations observed around gene poor loci.”

Supplemental Figure S4 - The “most clonal” integrations are shown (11 of them, it seems). How were these selected? This does not appear to be the top 50, which is what was used for many of the other displays. Ideally there should be some indication of the cutoff and/or another plot showing the lower clonality sites for this replicate.

Integrations could be either sorted by the clonality in each sample or by the average of clonality across samples. To avoid making this choice, we have replaced this figure with a detailed analysis of 6 replicate libraries prepared from each of two spleen DNA samples from the cohort. In one sample there are 4 clonal integrations found in 6 libraries and 1 clonal integration found in only 1 library. In the other sample there are more than 20 clonal integrations found in all libraries but also a number of clonal integrations found in only 1/6 libraries. The statistics of this overlap are summarized in Supplemental Figure 4. The take home message is that quantitation and ranking of clonal integrations is reproducible and that $NC > 0.1$ is a reasonable cutoff such that >95% of integrations will be found in all replicate libraries. Integrations with $NC < 0.1$ vary considerably in how often they are found in replicates however there is a clear trend between the number of libraries an insert is found in and its clonality.

Minor issues

Strand bias - when referring to a bias for the forward strand, is this in reference to the strand

of the gene to which the event was assigned or the strand of the chromosome?

We've clarified this in the text on p.10

“A recent study of MuLV induced T- cell lymphomas used orientation of integrations as evidence that there is selection for deregulation of nearby genes ⁵ i.e. loci bearing insertions that are decidedly biased in one direction are likely to have undergone selection for their effects on a nearby gene that would differ if the insertion orientation were opposite.”

Figure S3 could be improved for clarity. The last few steps of the sequencing (i.e. reading the barcodes) are somewhat irrelevant and don't add anything the figure unless I have missed something. It would be better if the focus of this figure was on how the method differs from the published procedure from which it was derived.

We have revised this figure to include a detailed comparison of previous library construction methods vs our new method. We include sequencing and barcode reading steps to make it easier for non-experts to understand the methodology which differs from construction of conventional NGS libraries. The final steps are also included because the orientation and ordering of sequencing reads has implications for cluster generation i.e. the first read cannot be 100% uniform. This motivated the strand switch such that the LTR is now sequenced from read 2.

*Figure S7 - why are these not symmetrical along the diagonal?
This analysis could be quite informative but the result is not readily interpretable as presented. Without knowing if/how P values were corrected it is unclear how much of what is seen as green is worth considering. A table ranked by pairs of loci might be superior.*

The figure (now Fig. S10) was only intended as an example (with others and corresponding tables on the website) but we have now included a table of the commutated pairs for all cohorts (background, genotypes, B/T) and this table includes multiple testing correction. (pp. 18-19 Supplemental Table 11).

Regarding symmetry, the exact tests used to calculate the picture are not identical across the diagonal. In one direction samples with/without a clonal mutation in window X have all their inserts in/out of window Y counted. In the other direction samples with/without a clonal mutation in window Y have all their inserts in/out of window X counted. This can be informative in some samples where there are multiple inserts within a single window. For instance, mutation of one gene early on and another gene later on can give a clonal insert at one locus and multiple subclonal inserts at the other.

The heat maps that are largely black with green, blue etc in the foreground could benefit from a different colour scheme. For example, white background and a Brewer palette representing the gradient of P values spanning significance levels.

We have trialed several schemes over the years and eventually settled on this one. However, in future we will take into account the growing popularity of Brewer palettes. Ultimately the values underlying the image are available in supplemental tables S2 should the reader be interested in particular locus.

Spelling error in legend for Table S7 - lymphoma

This has been corrected.

Why was this comparison only done using published copy number data? Since many of the integrations in this study were mapped to specific genes, it seems that the genome and exome studies would also be worth comparing to. There are several such studies. They could also be used to obtain a superior list of genes beyond those in the census for some of the comparisons discussed above.

We had prioritized copy number analyses in part because MuLV tends to deregulate intact ORFs and only occasionally creates truncations, and never SNVs. Nonetheless this is a worthwhile suggestion and we have now included a new comparison to all lymphoid malignancy datasets available from the cBio portal and found a significant overlap with our candidate gene lists and all cohorts. 78 genes were found mutated twice or more in the entire set of cohorts. This analysis has now been made Figure 6 and supplemental Table S7 and discussed pp. 13-14.

REVIEWERS' COMMENTS:

Reviewer #1 (Remarks to the Author):

The authors have sufficiently addressed all the concerns raised during the initial review. The manuscript is acceptable for publication, in my opinion.

Reviewer #2 (Remarks to the Author):

The majority of this referees comments were rebutted without substantive revision. There is not a systematic analysis of new driver genes rather than reporting of those confirmed in the literature. They decline to perform copy number analysis despite showing that this does adds data from the study described, and also as they have compared their results to human copy number data. The authors have now backed down from calling the serial samples "preleukemic" and it is now now clear what the serial samples thus reflect (Apart from ongoing mutagenesis). The correlation with RNA-seq doesn't appear to have been informative, or at least fully explored, to really verify the effect of mutations.

Reviewer #3 (Remarks to the Author):

The manuscript has substantially improved in this revision. I feel that all my original concerns have been adequately addressed.

REVIEWERS' COMMENTS:

Reviewer #1 (Remarks to the Author):

The authors have sufficiently addressed all the concerns raised during the initial review. The manuscript is acceptable for publication, in my opinion.

Reviewer #2 (Remarks to the Author):

The majority of this referees comments were rebutted without substantive revision. There is not a systematic analysis of new driver genes rather than reporting of those confirmed in the literature.

We have altered the title and abstract to clarify the purpose and findings of the study. The literature and growing body of exome studies are full of weakly evidenced lymphoid malignancy driver candidates (many are listed in Supplementary Table 9). Our study examines the extent to which subclonal mutations can be usefully inform which candidates with sparse evidence of clonal lesions should be prioritized for further study. The listing of known driver genes/loci is intended as validation of the approach. Many of the loci found in the comparisons with exome studies are evidenced by scant mutations and we have listed these in Figure 6 and Supplementary Table 7.

They decline to perform copy number analysis despite showing that this does adds data from the study described, and also as they have compared their results to human copy number data.

After examining previously published examples of copy number analysis of MuLV lymphomas we felt the yield would not add to the study except to re-establish the prior finding that they are relatively karyotypically stable compared with other models.

The authors have now backed down from calling the serial samples "preleukemic" and it is now now clear what the serial samples thus reflect (Apart from ongoing mutagenesis).

This was a useful suggestion. Other readers had also interpreted our description of the time course and use of the word preleukemic in the same manner as reviewer 2 and this has been clarified.

The correlation with RNA-seq doesn't appear to have been informative, or at least fully explored, to really verify the effect of mutations.

RNA-seq on all samples (rather than just the 26 included) would likely yield more genes deregulated by clonal integrations. At minimum these samples did demonstrate that mutations with a normalized clonality below 0.1 should not be expected to give significant changes in expression. Incidentally we were also surprised to find fusion transcripts are far rarer than might be expected based on previous studies of individual genes.

Reviewer #3 (Remarks to the Author):

The manuscript has substantially improved in this revision. I feel that all my original concerns have been adequately addressed.

NCOMMS-17-00438A-Z

'Tracking Subclonal Mutation Frequencies Throughout Lymphomagenesis Identifies Cancer Drivers in Mouse Models of Lymphoma.' Webster *et al.*

Reviewers' comments:

Reviewer #1 (Remarks to the Author):

This manuscript describes a variety of MuLV-induced models of leukemia/lymphoma in which insertional mutagenesis is monitored over a long-time period. The data offer a number of interesting results that pose a variety of hypotheses. However, I am not convinced yet by the quantitative claims made by the authors regarding their detection of specific viral insertion sites in complex samples. In addition, many of the analyses seem superficial in nature.

We appreciate the concerns regarding our quantitative claims and have now tried to better explain the rationale for and methodology of insert quantitation. We have included additional data and analyses to verify the reproducibility of our quantitation. We have also addressed several oversights in the comparative analyses of our data with human cancer genomes, which were pointed out by the reviewer, and we included new comparisons to human tumor genomes and RNAseq analysis of a subset of tumors.

Major comments

1) Figure S4a shows a sample that has insertion events with relatively high clonality scores (only 1 is < 0.6). How does the quantification look for insertion events that are less abundant in the sample?

The raw and normalized clonality values of inserts are reproducible only for clonal inserts and as expected, become noisier when inserts are represented by a single DNA fragment/read/cell. To explore this further we have analysed data from 2 series of 6 replicate libraries taken from 2 spleen DNA samples. The results from these replicate libraries are now included as supplementary figures S4a-e. These replicate series indicate that inserts that have a normalized clonality of 0.1 or above are found with a high degree of quantitative reproducibility whereas the vast majority of reads at lower clonalities, including the majority of single read integrations are only rarely found in more than one replicate. We have mentioned this analysis in the text (pp.8-9) and added a detailed explanation of this comparison in the legends of supplementary figure S4a-e.

Did the authors have repeated library preps for samples in the dilution series for the 9:1 or 99:1? This would be informative to see here.

We think that the repeat libraries discussed above can address this concern. The dilution experiment included 5 mixed samples where the clonal inserts from both DNAs behave as expected and proportions of the inserts within each sample remain consistent between samples. This demonstrates that read/fragment abundance is proportional to dilution with a slight bias for the lower clonality inserts and against the higher clonality ones.

2) I cannot read the axis labels for Figure 2c, but I assume that the maximum value on the y-axis is 1.0. Given that, it would appear that NC values of many of the insertion sites from Day 9 and Day 14 samples are >0.5. I would disagree with the authors that these insertion events are "sub clonal" as they assert. Perhaps this result is being skewed by the NC calculation. It

might be helpful to see a normalized read number from the HiSeq for this panel instead. If the read numbers for the insertion sites from the early time points are one or two orders of magnitude lower than the tumor samples, then the authors may want to reconsider how the data in Figure 2c are depicted.

It is true that presenting the normalized clonality values without non-normalized values could be easily interpreted as premalignant samples appearing to have clonal insertions. We have added plots without normalization to show that the number of fragments for the most abundant inserts is a much lower fraction of the sample in early samples than in late samples. We've also altered these figures to enlarge the axes markers and explained the y and x axes more clearly in the legend.

3) I do not understand Figure 4a as the legend and the graphic are not consistent. The legend states that the values are normalized to 1 while the graphic shows a scale up to 3.0. Does this mean that the relative abundance of insertions at some loci is 3 times greater in early time points than it is in tumor samples? This isn't clear.

We apologize for the confusion; the reviewer interpreted the data correctly, and we have now clarified this in the legend (this figure is now figure S5). For each window in the time series all the time points were normalized such that the final late stage disease time points are all set to 1, i.e. we are comparing how all top 50 clonal CIS windows progress over the time course. Looking at windows around individual loci, it is clear that some stochastic noise is present over the time course. Certain loci may also be selected against (the example of Smyd3 is discussed on p. 18 in the results/discussion). The global trend is clearer when viewing the median and mean values in Fig S5b) and S5c) but we felt it was important to show the values and stochasticity of individual loci as well.

4) I do not understand the main point of Figure 4d. The distinction between late-stage clonal and late-stage sub-clonal events depends entirely on the performance of the sequencing method employed to identify the insertion sites. I am not convinced by the data in Figure S4 that the method has sufficient sensitivity to reliably distinguish clonal from sub-clonal events in tumors. Without additional data showing the performance of the method for insertions in the "sub-clonal" range, I think the authors cannot make significant conclusions regarding clonal vs sub-clonal events in tumors. Moreover, I do not think that this distinction needs to even be made for the major conclusions of the manuscript. Instead I think it's an unnecessary distraction. I would suggest that the authors merge the "late clonal" and "late sub-clonal" data sets into a single "late event" data set.

At the onset of the study, we could not say it was justified to include all subclonal/single read mutations in the analysis and felt that treating all mutations as equivalent would be discarding (admittedly partial) information about mutation abundance. Our best estimate was that of the of 700,000 mutations in the data set only 3,000 were clonal. We felt it was important to demonstrate as thoroughly as possible whether low clonality integrations should be excluded or if their inclusion would be useful and could be justified. The main point of Figure 4d (now Figure S5d) was to demonstrate that within the top 50 CIS loci, subclonal and clonal events perform similarly. We are not attempting a precise distinction between clonal and subclonal events, but rather would like to show that subclonal events are informative. We have now moved this analysis to the supplementary material to avoid distracting the reader. After having justified the use of all inserts in this manner, the subsequent analyses in the manuscript use pooled late stage data, as suggested by the referee.

5) What is the significance of the overlap described between the author's data set and the GWAS studied cited? some overlap is expected simply due to chance. Is the observed degree of overlap much greater than expected? A simple Fisher's Exact test, or something comparable, should be reported. If the results are not significant, this section should be removed.

To place a p-value on this analysis is less trivial than it seems. This is due to the highly variable and overlapping scope of most lymphoid malignancy GWAS studies (some cohorts have been extended and republished many times and generally they find the same loci). Nonetheless having performed an extensive review and summary of lymphoid malignancy GWAS cohorts, we have taken all disease SNPs identified in a series of 18 studies published since 2009 (supplemental table S10), mapped these SNPs over to the mouse genome and shown significant overlap of our loci in a candidate gene independent manner (using Fisher's exact test of the 26 GWAS loci of combined studies we find 14 correspond to our CIS, the expected number by chance would be 0.82).

6) The results of the gene ontology analysis need to be paired down. The authors currently report over 5000 GO terms in the supplemental table S8. Over 1000 of these have an FDR < 0.05. This draws into question the relevancy of the results that are highlighted in the manuscript. With such a large number of hits, how meaningful is it that these specific categories appear on this list?

We appreciate the reviewer's point and have removed this table in the current version since we don't feel it adds a great deal to the analysis and was a token inclusion for descriptive purposes "Gene ontology analysis of the entire list of candidates yields search terms that are dominated by lymphocyte biology, notably lymphocyte activation, lymphocyte aggregation and cell-cell adhesion" We felt this sentence represented the gist of the top 30 terms in the list.

The authors use these results to suggest that classical and non-classical MHC pathways are involved in lymphoma progression. It is unclear why these pathways are emphasized while hundreds more go without mention. Moreover, the authors fail to acknowledge that their leukemia/lymphoma model is virally-induced. Prior work has implicated MHC in the immune response to MuLV. Given this prior work, what is the evidence that the MHC connection in this model is relevant for human disease that is not virally-induced?

We felt it worthwhile to discuss costimulatory molecules and MHC loci since they were more prevalent in this study than prior insertional mutagenesis studies. We've now included mention of these on p.17, which is arguably a better place to discuss them since these loci tend to be BCL2 and B cell specific and we cite corroborating evidence from human studies and GWAS of non-viral mature B lymphoid malignancies. It is certainly true, as the reviewer pointed out, that some mutations of immune mediator/modulator genes might also be selected as a consequence of virus host interactions and we have now explicitly mentioned this caveat on p.17.

"Whilst loss of MHC loci (or lowered expression) in human B malignancies likely results from selection to escape anti-tumour immune responses, viruses that infect human lymphoid cells (including those that cause malignancies such as Epstein Barr Virus and Human T-cell leukaemia virus) encode mechanisms to avoid host immune responses. In the context of MuLV infection, mutations at MHC loci may therefore also represent selection against host antiviral responses rather than antitumor responses."

Minor comments

Figure 3e & f
Add labels to y-axis

Figure 4a-c
Add labels to y-axis

These figures have been modified as suggested. Figure 4 is now Figure S5.

Reviewer #2 (Remarks to the Author):

This manuscript describes the use of retroviral mutagenesis modeling to establish lymphoid malignancies in WT or BCL2 transgenic mice (to promote B lineage disease) with state of the art Illumina next generation sequencing to identify retroviral integrations. The authors profile the lineage and differentiation stage of the tumours and use two experimental and computational approaches to identify subclonal events that may then be described as driver lesions. The experimental approach is sacrificing cohorts of mice at early time points (that are deemed "preleukemic") and comparing these with mutations identified at later time points. The second is to use computational approaches - entropy and dynamic time warping - to infer these lesions. The authors spend some time focusing on the type of integration (E.g. strandedness) and specific sites (such as the well described MYC/PVT region).

The authors are experts in the use of these models, and the efforts to extract, model and exploit these subclonal data are impressive. I have relatively few concerns about the execution of the experiments, but I do have several major concerns about the design of the study, and whether it is able to definitively address the notion that subclonal lesions are drivers.

My greatest concern is intra-animal tumor heterogeneity - that is that the majority of animals clearly have at least 2 tumors of different lineage. This suggests that there are multiple genetically independent tumor populations - probably many more - and that if there are subclonal lesions that change over time, they reflect one distinct tumor winning out over time, rather than reflecting human disease, where multiple clones from a founding ancestral clone are propagated, and subclonal genetic lesions can emerge over time.

We appreciate these concerns and think they might arise at least in part from lack of clarity in how we described our aims and the model used. Our study is not intended as an equivalent to exome resequencing tumor lineage studies from multiple biopsies of human tumors. It is rather an analysis of the entire mutational spectrum of malignant clonal outgrowths in parallel to subclonal and premalignant/non-malignant tissues. As such we have:

- Taken the most clonal mutations as a gold standard for this model.
- Compared this loci list to known human cancer genes.
- Expanded analysis to include subclonal/low abundance mutations (that outnumber clonal mutations by >100 to 1).
- Demonstrated the additional usage of the subclonal identifies additional loci that are also human cancer genes.

There is no real human equivalent to this study because the sensitivity of insertion mutation cloning as has no real equivalents in human cohorts. We demonstrate that low abundance mutation selection is observed at loci that only rarely contribute to malignant clonal outgrowth and as such can be used to strengthen the case for selection of driver mutations

that only rarely achieve clonal outgrowth.

We also now explicitly state the nature of MuLV disease in the text.

“A single clonal outgrowth of pure tumor cells containing few mutations will yield high coverage of each mutation, whereas a clonal outgrowth with dozens of concurrent mutations alongside a large proportion of non-tumor DNA will yield low coverage for even the most clonal mutation. Additionally, there may be multiple independent clones of varying clonality within each tumour and multiple independent subclones of each lymphoma.”

Mouse models of cancer with 100% penetrance must statistically have multiple competing premalignant/malignant clones at different stages of development. In the human scenario, the extent to which driver mutations are also represented in premalignant clones/non-tumor is unknown for all but a few genes, but we agree they are unlikely to be as abundant as seen in our model. Nevertheless, we feel the comparison and joint analysis of all mutations from both malignant and premalignant cells can greatly increase the confidence of genes identified as drivers of clonal selection. Although we can't distinguish between the subclonal mutations present in the dominant tumor clone vs single bystander lymphocytes, we can say that aggregate analysis of both is informative.

This multi-tumor composition also affects analysis and interpretation. If there are at least two completely distinct tumors (one B, one T for example) they may have "clonal" lesions specific to each lineage of tumor that the current approach may misinterpret as subclonal (drivers). Such lesions can then become predominant if one tumor simply outgrows another. This may lead to the erroneous conclusion that these "subclonal" lesions are drivers - they may have been clonal in that tumor throughout. Sorting the tumor populations into each lineage, and/or clonality analysis, and/or single cell analysis would be required to resolve this.

We would like to clarify that the vast majority of our mutations are indeed not driver mutations. We merely demonstrate that in small scale biopsies (averaging 1-2mm in size) there are enough subclonal events corresponding to known driver mutations that the data can be used to supplement analysis of clonal mutations i.e. the findings of both datasets overlap sufficiently to justify the use of both. We have tried to avoid stating that these mutations are “subclonal drivers” but rather would say that subclonal mutations are indicative of driver status often enough to be useful data. We don't feel this is a given.

Regarding separation of lineages or clones, sorting by immunophenotype doesn't allow separation of independent clones within lineages. For a small number of cases we have separated B & T cell populations prior to insert cloning but found that the insert cloning method is so sensitive that even “pure” populations contain a fraction of the inserts of the other lineage. This is an inherent caveat of highly sensitive PCR based methodologies. We do observe percentages of B & T lineages and mutation frequencies correlate with these, but we have avoided overinterpreting these associations.

It is also unclear if the serial samples address the questions posed. These do not reflect the evolution of a tumor over time, but a sampling of a cohort at different time points. The authors also state that these are preleukemic - presumably as the animals are not moribund - but this doesn't mean they are not preleukemic - one would have to purify the populations and perform secondary transplants to address this.

The time course experiment is used to compare mutation frequencies from the early/late time points; it is not an attempt at lineage tracing within individual animals. The early/late

comparision successfully identifies known cancer driver loci and demonstrates that strand bias increases at later points in tumor development, thus further validating our approach. We cannot make claims about mutation order within each sample, but simply observe abundance of mutations at different loci in populations of different ages.

For want of a better term, we had referred to premalignant tissue/samples/mutations, but whenever possible, we have now changed these terms to “early stage” or “late stage” tissue and mutations. It is correct that we do not know if all biopsies taken from “pre-malignant” animals contain cells that will eventually become malignant, we can only say that the disease is 100% penetrant.

The data presented are limited to presentation of integration sites only. There is no consideration of other modalities of genomic alteration - DNA copy number alterations, sequence variations - that do arise in such screens and may impact the lanscape of alterations described, and may also influence clonality analysis (e.g. see Dang et al Blood 2015).

The study mentioned (Dang et al. <https://doi.org/10.1182/blood-2015-02-626127>) is one of the better examples of why we decided against multimodal analyses. This study finds an average of 25 nonsynonymous coding mutations per MuLV tumour (n=10) which is fivefold less than found in their ENU cohort. Within the MuLV tumours, only 15 genes were recurrently mutated. The CGH of this study of the 3 MuLV tumors analyzed only one copy number change (trisomy 15) was observed in one sample. In the entire study, copy number aberrations were sparse and generally represented events that had been previously described e.g. loss of *Ikzf1*, *Nf1*, *Cdkn2a/b*.

Other studies with MuLV and other insertional mutagens have found little or no copy number changes and very little exonic sequence variation primarily because these models are driven by deregulation of ORFs (equivalent to translocations and/or non-coding mutations) and occasionally intragenic mutations (which can be truncating/inactivating/stabilizing). We could have included similar analyses, but the yield of new findings would arguably be minimal compared to the cost. We find that more benefit is gained from comparisons to human tumor datasets.

A concern is inference of clonality from the sequencing data. There is apparently no stringency at all - as few as 1 read is called as an integration. The method is also unclear. For variant calls, one would compare mutant v germline allele fraction.

We have better illustrated the method in a revised Figure S3. Our insert mutations are identified based on two nested rounds of ligation mediated PCR and a further nested sequencing primer. We are only amplifying mutant DNA and are therefore unable to assess the mutant/germline fraction. The method is a tradeoff between sensitivity (nearly everything sequenced represents a mutation) vs quantitation. The replicate samples demonstrate that the method is capable of reproducibly finding the most clonal insertions and these are also identified by whole genome sequencing. Although single reads of SNVs cannot be distinguished from sequencing errors, single reads of translocations/insertions/deletions can be readily distinguished from sequencing error and the only grounds for excluding them is if they are artefacts of library construction.

We have also resequenced some of our libraries using two primers that are set further back from the virus genome junction and have found that between 95-97% of read pairs with read 2 phred score cutoff of >30 correspond to a virus genome junction. Low quality read 2 presumably stems from an inability of the read 2 primer to bind the library due to lack of an

LTR sequence i.e. a PCR product that doesn't contain the end of the LTR cannot be sequenced. The small number of high quality read 2 that don't contain a virus genome junction appear to correspond to ligation and PCR artifacts that either fail to map to the mouse genome or are removed by filtering steps during final processing of the data.

It is ambitious to use all inserts including those represented by a single read, and this is why we went to such lengths to justify the use of all mutations, but readers who would like to use the dataset for subsequent analyses are able to filter according to their preferred read depth or clonality.

As a minor point, it is not clear why human and mouse normal DNA were sequenced.

The uninfected mouse DNA and human DNA is sequenced to identify any PCR artifacts that might be attributable to cross contamination of reagents/adjacent wells (which would be found in human DNA) or mispriming events on the mouse genome (there are about 18 endogenous retroviral loci that loosely match our primer sets within the mouse germline). We now explain the use of control DNAs in the supplemental methods section pp. 4-5.

Here the normalization appears to be to the sheared fragments - but wouldn't this be genome wide, not a site-specific analysis? Many other factors including adapter ligation efficiency, insert size, PCR amplification efficiency, clustering efficiency etc. would have an effect on the N of reads per site. It is not clear that the dilution series of a clonal variant addresses this.

We don't wish to oversell the quantitation other than to say it is reproducible (now better illustrated in supplemental Fig. S4) and demonstrably concentration dependent in a dilution series i.e. the same mutation will be sequenced with roughly 100-fold less coverage if it is 100-fold less abundant. This does not eliminate differences between loci and there will certainly be blind spots in the genome where some inserts will be underrepresented or missed but this is true of any genome wide study. We have added a paragraph to the supplemental methods section (pp. 4-5) that outlines the greatest potential sources of error in our protocol.

*"Several potential sources of error may influence quantitation based on fragment counts. The EcoRV digestion step (designed to remove sequences that run into the virus from the 3' LTR) means approximately 5% of the genome within 150 bases of an EcoRV site will be underrepresented or eliminated. Sequence dependent amplification bias (e.g. GC rich regions being underrepresented) is minimized by cutting the number of PCR cycles down from a total of over 50 in prior protocols to 25. Furthermore, some inserts corresponding to highly recurrent sites (such as the 3' UTR of *Mycn*) may be eliminated in the final filtering stage, because they can be mistaken for recurrent PCR artifacts and/or cross contamination."*

If the analysis is to be restricted to RIS analysis, this is not developed enough. The inference is based on gene name and proximity, with no consideration of effect (Activation? inactivation?) of the insertion; RNA_seq of a subset of samples would address this.

We employed the automated prediction of targets using the KCRBM software package that was originally trained on a set of MuLV lymphoma expression arrays. We have now included the list of most likely mechanisms of nearby gene deregulation that are generated by this software package in table S2.

As mentioned above we have now also performed RNAseq on 26 samples. Subclonal integrations would not be expected to yield significant alterations in expression. Within this

cohort there are too few samples with clonal integrations at any given locus to demonstrate statistical significance for deregulated expression. Nonetheless by looking for modified transcripts we find a handful of expected fusion transcripts of *Mycn* and *Notch1* (as described in previous screens) and the degree of clonality corresponds with the abundance of these transcripts.

The ability to interpret and have confidence in the authors conclusions is limited by the difficulty in understanding what the new algorithms achieve - the Entropy and Dynamic Time Warping analyses.

We have now explained this more clearly. The entropy approach we used to stage tumors is similar to the Shannon entropy approach used by *Brown et al.* and *Baldow et al.* (references 13 & 14). Nonetheless we wanted to employ two independent methods for staging samples. To this end we employed a novel approach to cluster based on the use of distance measures (specifically using dynamic time warp). DTW is a well-accepted methodology for comparison of ordered data series. We have also now used the Kolmogorov Smirnov statistic and obtained nearly identical groupings.

It would be of interest to have a clearer presentation of new genes (if any) identified by this study. Many (e.g. SFig 5) appear to be well credentialled genes.

The genes shown were chosen to demonstrate that subclonal events are indicative of the drivers found in human data. The definition of “new” genes is somewhat fuzzy. Many of the genes listed in table S5 are from single studies that require further confirmation. The remaining loci from our study have no prior evidence we could find in the literature or are evidenced in the newly added comparison to human exome studies of lymphoid malignancies (included at the request of reviewer 3).

We have now combined and compared our screen with human copy number, GWAS and exome studies as well as curated lists of cancer genes. Cancer genome datasets invariably generate a few unequivocal drivers and a much longer list of weaker candidate genes, whose functional relevance can only be established by integration with other datasets or follow up studies. Our study generates hundreds of loci that survive correction for multiple testing. More than 100 of these have precedent in the literature and we feel the remainder are a valuable resource for prioritizing the study of other candidates.

Reviewer #3 (Remarks to the Author):

Tracking Subclonal Mutation Frequencies Throughout Lymphomagenesis Identifies Cancer Drivers in Mouse Models of Lymphoma

This manuscript describes the use of the murine leukemia virus to catalogue the integration sites genome-wide that demonstrate patterns indicative of driving lymphomagenesis. The authors employ some novel approaches to map integration sites and deconvolute clonal from sub-clonal selection and several transgenic backgrounds (Vav-BCL2, Emu-BCL2, wild-type) to aid in promoting B-cell lymphomas. The authors characterized ~500 induced B-cell lymphomas and T-cell ALLs using their integration site sequencing strategy that allowed detailed characterization of the evolutionary/selection history in each mouse. The integrations are examined with respect to strand bias, which can indicate a preference towards events that promote expression of nearby genes.

In general, this work promotes an increased statistical power in their somewhat novel design that was achieved by leveraging the signature of selection for subclonal as well as clonally-selected integrations among the premalignant and malignant cells, respectively. This is framed as a key feature of the study that sets it apart from previous attempts to map cancer drivers using viral integration screens. Their strategy is very clearly supported by the recapitulation of an extensive set of genes orthologous to known human cancer genes including many relevant to B-cell lymphomas. This is an interesting and novel study that yields a potentially invaluable resource for researchers who study the genetic nature of lymphomas.

Major issues

Comparison to the cancer gene census is central to many of the displays and tables. I am worried the authors failed to ensure whether there are recently described lymphoma-related genes absent from the census. For example, MEF2B, GNA13, S1PR2 are fairly well established lymphoma genes that are lacking from the CGS. There are quite likely others missing as well. It seems reasonable to add a more curated set of genes that are relevant to the diseases being studied here.

This is a good suggestion and we have now included a new analysis comparing our data to exomes of several lymphoid malignancy cohorts and found significant overlap (Figure 6 pp.13-14): 78 MuLV candidate genes were found mutated in 2 or more samples of these studies.

I was unfamiliar with the dynamic time warping methodology and, although the application seems valid, it appears to (potentially) be an unnecessary complication to the paper. In Figure 3, it seems that the Shannon entropy distribution plotted as a histogram would have an obvious cut point around 3.5. This is only shown as boxplots (panel C) after the two groups were defined and the distribution (by rank), in panel B so it is not possible to tell from the data whether the approach used to make panel A was actually needed. Further, even if the entropy score alone does not capture enough of the information from each mouse, it seems that a Spearman correlation would be applicable here since the integrations are being described by rank already.

We appreciate that the use of DTW in this context is novel, but we included it because it is an entirely independent approach that yields a similar result to the use of entropy scores. While the use of the Spearman correlation is not entirely appropriate here (because it is a rank-based test), based on this suggestion we have now also used the Kolmogorov Smirnov statistic as another distance measure, which gave nearly identical groupings as obtained by DTW.

P values are used in quite a few of the tables and display items but the statistical approaches used to derive them are not explained in sufficient detail. Here are some examples: Contingency table tests used for determining significant integration sites. A formal explanation of how these are performed using these data should be included in the Methods. The same goes for the contingency tests used to generate Figure S7. Furthermore, were the P values corrected for the many tests that were performed? If so, how?

We apologize for not providing sufficient detail and have now tried to make all statistical methodologies more apparent to the reader. The statistics of our early/late, genotype, cell type and strand bias were only described in the legends of supplementary tables, but we have now clarified this in the main text. All contingency table tests were Fisher's exact test and whilst the methodology is explained in the legends for supplementary tables we have now added it to the Methods sections and clarified this in the text. Figure S7 has been

updated to include multiple testing correction in the tabulated version of results. In this analysis dozens of commutated gene pairs survive when analyzing the larger cohorts, which would not be the case were the study limited to analysis of clonal integrations.

Supplemental Figure S6 is a very useful resource as I'm sure the browser will be. There is an unsatisfying paucity of integration with any genomic annotations beyond genes and the cancer census. These figures indeed raise some interesting trends that are worth some discussion in the text. PRDM1, for example, has recurrent integrations spanning an extremely large region. It is difficult to discern how these can all be impacting PRDM1 expression. The REL locus is also quite interesting because there appears to be two regions that are enriched for integrations (upstream of REL proper and downstream of BCL11A). This is suggestive of distal regulatory elements that may be enhanced by these integrations. Are there existing data sets for mouse that would reveal whether there is coincidence of integration sites and potential regions that act as enhancers in mature B cells? If such data exist, it would be worthwhile to discuss this in the context of the present data.

Mapping MuLV integrations with respect to genomic features such as promoters, expressed genes and enhancers has an extensive literature in its own right and this is why we have gone to such lengths to differentiate between unselected and selected integrations by multiple criteria without sole reliance on density. The potential for retrovirus integrations to act over large distances has been documented including a very thorough recent study by Babaei *et al.* (<https://www.nature.com/articles/ncomms7381> ref 28). Whether allocation of candidate genes is automated or curated will impact the results, but this is also true of cancer studies of GWAS, epigenomes and copy number variation. We have now elaborated on these issues in the results (p. 15).

“MuLV is known to deregulate genes via a diverse set of mechanisms including distal interactions of virus enhancers via 3D conformation of the genome²⁸. The integration bias of MuLV for enhancers²⁹ and capacity for long range interactions likely explains the clusters of late stage/strand biased/genotype specific integrations observed around gene poor loci.”

Supplemental Figure S4 - The “most clonal” integrations are shown (11 of them, it seems). How were these selected? This does not appear to be the top 50, which is what was used for many of the other displays. Ideally there should be some indication of the cutoff and/or another plot showing the lower clonality sites for this replicate.

Integrations could be either sorted by the clonality in each sample or by the average of clonality across samples. To avoid making this choice, we have replaced this figure with a detailed analysis of 6 replicate libraries prepared from each of two spleen DNA samples from the cohort. In one sample there are 4 clonal integrations found in 6 libraries and 1 clonal integration found in only 1 library. In the other sample there are more than 20 clonal integrations found in all libraries but also a number of clonal integrations found in only 1/6 libraries. The statistics of this overlap are summarized in Supplemental Figure 4. The take home message is that quantitation and ranking of clonal integrations is reproducible and that $NC > 0.1$ is a reasonable cutoff such that $>95\%$ of integrations will be found in all replicate libraries. Integrations with $NC < 0.1$ vary considerably in how often they are found in replicates however there is a clear trend between the number of libraries an insert is found in and its clonality.

Minor issues

Strand bias - when referring to a bias for the forward strand, is this in reference to the strand

of the gene to which the event was assigned or the strand of the chromosome?

We've clarified this in the text on p.10

"A recent study of MuLV induced T- cell lymphomas used orientation of integrations as evidence that there is selection for deregulation of nearby genes⁵ i.e. loci bearing insertions that are decidedly biased in one direction are likely to have undergone selection for their effects on a nearby gene that would differ if the insertion orientation were opposite."

Figure S3 could be improved for clarity. The last few steps of the sequencing (i.e. reading the barcodes) are somewhat irrelevant and don't add anything the figure unless I have missed something. It would be better if the focus of this figure was on how the method differs from the published procedure from which it was derived.

We have revised this figure to include a detailed comparison of previous library construction methods vs our new method. We include sequencing and barcode reading steps to make it easier for non-experts to understand the methodology which differs from construction of conventional NGS libraries. The final steps are also included because the orientation and ordering of sequencing reads has implications for cluster generation i.e. the first read cannot be 100% uniform. This motivated the strand switch such that the LTR is now sequenced from read 2.

*Figure S7 - why are these not symmetrical along the diagonal?
This analysis could be quite informative but the result is not readily interpretable as presented. Without knowing if/how P values were corrected it is unclear how much of what is seen as green is worth considering. A table ranked by pairs of loci might be superior.*

The figure (now Fig. S10) was only intended as an example (with others and corresponding tables on the website) but we have now included a table of the commutated pairs for all cohorts (background, genotypes, B/T) and this table includes multiple testing correction. (pp. 18-19 Supplemental Table 11).

Regarding symmetry, the exact tests used to calculate the picture are not identical across the diagonal. In one direction samples with/without a clonal mutation in window X have all their inserts in/out of window Y counted. In the other direction samples with/without a clonal mutation in window Y have all their inserts in/out of window X counted. This can be informative in some samples where there are multiple inserts within a single window. For instance, mutation of one gene early on and another gene later on can give a clonal insert at one locus and multiple subclonal inserts at the other.

The heat maps that are largely black with green, blue etc in the foreground could benefit from a different colour scheme. For example, white background and a Brewer palette representing the gradient of P values spanning significance levels.

We have trialed several schemes over the years and eventually settled on this one. However, in future we will take into account the growing popularity of Brewer palettes. Ultimately the values underlying the image are available in supplemental tables S2 should the reader be interested in particular locus.

Spelling error in legend for Table S7 - lymphoma

This has been corrected.

Why was this comparison only done using published copy number data? Since many of the integrations in this study were mapped to specific genes, it seems that the genome and exome studies would also be worth comparing to. There are several such studies. They could also be used to obtain a superior list of genes beyond those in the census for some of the comparisons discussed above.

We had prioritized copy number analyses in part because MuLV tends to deregulate intact ORFs and only occasionally creates truncations, and never SNVs. Nonetheless this is a worthwhile suggestion and we have now included a new comparison to all lymphoid malignancy datasets available from the cBio portal and found a significant overlap with our candidate gene lists and all cohorts. 78 genes were found mutated twice or more in the entire set of cohorts. This analysis has now been made Figure 6 and supplemental Table S7 and discussed pp. 13-14.